# Approximations to worst-case data dropping: unmasking failure modes

**Jenny Y. Huang**                                                          *jhuang9@mit.edu*
*MIT EECS and MIT-IBM Watson AI Lab*

**David R. Burt**                                                           *dburt@mit.edu*
*MIT EECS and MIT-IBM Watson AI Lab*

**Yunyi Shen**                                                              *yshen99@mit.edu*
*MIT EECS and MIT-IBM Watson AI Lab*

**Tin D. Nguyen**                                                           *tdn@mit.edu*
*MIT EECS and MIT-IBM Watson AI Lab*

**Tamara Broderick**                                                        *tbroderick@mit.edu*
*MIT EECS and MIT-IBM Watson AI Lab*

**Reviewed on OpenReview:** *https://openreview.net/forum?id=m6EQ6YdPXV*

## Abstract

A data analyst might worry about generalization if dropping a very small fraction of data points from a study could change its substantive conclusions. Checking this non-robustness directly poses a combinatorial optimization problem and is intractable even for simple models and moderate data sizes. Recently various authors have proposed a diverse set of approximations to detect this non-robustness. In the present work, we show that, even in a setting as simple as ordinary least squares (OLS) linear regression, many of these approximations can fail to detect (true) non-robustness in realistic data arrangements. We focus on OLS in the present work due its widespread use and since some approximations work only for OLS. Across our synthetic and real-world data sets, we find that a simple recursive greedy algorithm is the sole algorithm that does not fail any of our tests and also that it can be orders of magnitude faster to run than some competitors.

## 1   Introduction

Researchers typically run a data analysis with the goal of applying any conclusions in the future. For instance, economists run randomized controlled trials (RCTs) of microcredit with a particular set of people at a particular time. If the resulting data analysis shows that microcredit increases business profit, a policymaker might distribute microcredit to people in the future, on the assumption that microcredit will help these people too. We might worry whether this assumption is warranted if we could drop a very small fraction of people from the original trial and instead conclude that microcredit decreases business profit. As a concrete example, Broderick et al. (2020) show that it is possible to drop 15 households out of over 16,500 in an influential microcredit RCT and change the result to a statistically significant conclusion of the opposite sign.

In many cases, then, it behooves us to check: can we find a small fraction of data that, if dropped, would change the conclusion of the analysis? A brute force approach to answering this question enumerates every possible small data subset and re-runs the analysis a combinatorially large number of times. E.g., suppose we might be concerned if dropping 0.1% of our data could change our conclusions. Running $\binom{16500}{16}$ 1-second-long data analyses would take over $10^{46}$ years.

Given these computational challenges, researchers have suggested various approximations instead. Broderick et al. (2020) suggest using an approximation based on instantiating continuous weights on the data points and differentiating with respect to these weights. The authors use this approximation to identify small data subsets that, when dropped, change conclusions in multiple landmark papers in economics (e.g. Angelucci & De Giorgi, 2009; Finkelstein et al., 2012). In follow-up work focused on OLS, Kuschnig et al. (2021) introduced two additional ideas for finding the worst-case data subset: (1) approximating the impact of removing a group of points by the sum of the impacts of exactly removing individual data points and (2) greedily removing one data point at a time. Moitra & Rohatgi (2023); Freund & Hopkins (2023) provide additional approximations that are specific to OLS.

Recently, scientists and social scientists have used some of these approximations to assess the robustness of important findings in econometrics (Martinez, 2022), epidemiology (Di & Xu, 2022), and the social sciences (Davies et al., 2024; Burton & Roach, 2023). Given the deployment of these approximations in practice, we ask when and how they can fail in realistic data analyses—to alert practitioners and motivate further approximation development. Previous works have identified particular instances of failure modes, without a comparison of failures across approximation methods (Broderick et al., 2020; Nguyen et al., 2024). Other works have illustrated failure modes in adversarial constructions or settings where a large fraction of the data ($\geq 1\%$) needed to be removed (Moitra & Rohatgi, 2023; Freund & Hopkins, 2023; Kuschnig et al., 2021).

In the present work, we systematically explore whether approximations can detect if there exists a very small fraction ($< 1\%$) of data that, if dropped, can change conclusions. In other words, we ask whether approximations can detect a particular form of non-robustness in a data analysis. We focus on natural data settings with no adversary. In order to include the approximations of Moitra & Rohatgi (2023); Freund & Hopkins (2023) in our comparison, we focus on linear regression fit with OLS. Before the present work and contemporaneous work by Hu et al. (2024), there had not been studies systematically characterizing non-adversarial failure modes of approximations for this form of robustness or studies comparing the prominent existing approximations.

In many aspects, Hu et al. (2024) and our present work are complementary. While Hu et al. (2024) focus on exact recovery of the most influential data subset with cardinality at most equal to a stated value, we focus on finding whether there exists a small subset of data that, if dropped, could change substantive conclusions. Given our different focuses, Hu et al. (2024) find it useful to separate masking into two phenomena: amplification and cancellation. Meanwhile, we find it useful to point out failure modes due to poor conditioning of the bulk of data. Hu et al. (2024)'s theory assumes a particular data-generating process [their Equation 7]; while we don't make this assumption, we instead need to take a limit of an outlier data point's position to derive our results (Proposition 4.3).

Both Hu et al. (2024) and our work focus on OLS for theory and illustration of failure modes. While linear regression is less common in engineering disciplines, Castro Torres & Akbaritabar (2024) demonstrate that, as recently as 2022, (often well) over half of all papers reporting any methods in Medical and Health Sciences, Agricultural Sciences, Social Sciences, and the Humanities used linear regression. Indeed, most of the applied papers cited in the discussion above use linear regression (Angelucci & De Giorgi, 2009; Finkelstein et al., 2012; Martinez, 2022; Davies et al., 2024; Burton & Roach, 2023). We suspect OLS is the most common form of linear regression used in practice.

In the present work, we identify failures to detect (true) non-robustness in approximations from Broderick et al. (2020), Kuschnig et al. (2021), Moitra & Rohatgi (2023), and Freund & Hopkins (2023). We are able to identify failures even in linear regression with a single covariate and an intercept term. In contrast to the two works most similar to our own (Kuschnig et al., 2021; Hu et al., 2024), we compare to three new OLS-specific approximations developed by Moitra & Rohatgi (2023); Freund & Hopkins (2023), and we present a theoretical runtime analysis along with an empirical runtime comparison of all methods presented. Importantly, we present new, targeted illustrations of failure modes, each aimed at revealing a specific factor contributing to failure (e.g., a point with extremely high leverage and low residual, a small clump of points far away from the rest of the data).

Across worst-case data-dropping approximations, we conclude that a simple greedy algorithm (suggested by Kuschnig et al. (2021)) does not fail our accuracy tests (both on synthetic and real-world data), is

conceptually straightforward, and can offer orders of magnitude savings in running time over the OLS-specific mathematical programming alternatives.

Code for our work is available at `gradientBasedDataDroppingFailureModes`, including all scripts for reproducing the results in this paper.

## 2 Setup

We first establish notation for OLS analysis paired with worst-case data dropping. The approximations of Moitra & Rohatgi (2023); Freund & Hopkins (2023) require that the data analysis be linear regression fit with OLS, and moreover that the data-analysis conclusion be changed if the sign of a regression coefficient were to change. To include these methods in our comparison, we focus on this case.

In particular, let $N$ be the number of data points. We write the data as $d_{1:N} := \{d_n\}_{n=1}^N$, where $d_n := (x_n, y_n)$ consists of covariates in a column-vector $x_n \in \mathbb{R}^P$ and scalar response $y_n \in \mathbb{R}$. OLS estimates an unknown column-vector parameter $\theta \in \mathbb{R}^P$ by minimizing a sum of squared losses to a linear trend: $\hat{\theta} = \arg\min_\theta \sum_{n=1}^N (y_n - \theta^\top x_n)^2$. We will often (but not always) include an intercept term, in which case we think of the $P$th covariate as an all-ones covariate.

We focus on conclusions that would be changed if the direction (i.e., sign) of an estimated effect $\hat{\theta}_p$ changed. For example, in "Contradicted and Initially Stronger Effects in Highly Cited Clinical Research," Ioannidis (2005) describes two studies that initially concluded hormone therapy reduces coronary artery events in women—and also two larger studies that instead concluded hormone therapy *increases* coronary artery events; due to the change in sign, Ioannidis (2005) counts the first two studies as "contradicted findings." Similarly, in *Mostly Harmless Econometrics*, Angrist & Pischke (2009, end of Section 2.2) describe conflicting regression analyses that conclude standardized test scores increase or decrease, respectively, with class size. As in the examples above, the sign of an effect often guides interpretation and decision-making in fields such as biomedicine or economics. Indeed, (Gelman & Carlin, 2014) argue that Type S (sign) errors (and also Type M, for magnitude, errors) are more relevant for data analysis practice than conventional Type 1 and 2 errors.

We might be concerned if dropping a small fraction $\alpha \in (0, 1)$ of our data changed our substantive conclusions. The value of $\alpha$ is user-defined. We follow Broderick et al. (2020) and use $\alpha = 0.01$ (i.e., 1% of the data) as a default. Broderick et al. (2020) define the *Maximum Influence Perturbation* as the largest possible change induced in some quantity of interest by dropping at most $100\alpha\%$ of the data. Since we presently assume conclusions are made from the sign of $\theta_p$, our quantity of interest will always be $\theta_p$. Without loss of generality, we assume $\hat{\theta}_p > 0$, and we ask whether we can change the result to a negative sign.

To write the optimization problem implied by the Maximum Influence Perturbation (Equation (1) below), let $w_n$ represent a weight on the $n$th data point, and collect a vector of data weights, $w := (w_1, ..., w_N)$. Define $\hat{\theta}(w) := \arg\min_\theta \sum_{n=1}^N w_n (y_n - \theta^\top x_n)^2$. Setting $w = 1_N$, the all-ones vector of length $N$, recovers the original data analysis, and setting $w_n$ to zero corresponds to dropping the $n$th point. We collect all weightings that correspond to dropping at most $100\alpha\%$ of the data in $W_\alpha := \{w \in \{0, 1\}^N : \sum_{n=1}^N (1 - w_n) \le \alpha N\}$. Finally, the Maximum Influence Perturbation for this particular OLS effect-size quantity of interest[1] can be written

$$\max_{w \in W_\alpha} \left( \hat{\theta}_p(1_N) - \hat{\theta}_p(w) \right). \tag{1}$$

The *Most Influential Set* is defined to be the set of dropped data corresponding to the maximizing $w$ value.

In principle, one might solve Equation (1) by computing $\hat{\theta}_p(w)$ for each of the $\binom{N}{\lfloor \alpha N \rfloor}$ values within $W_\alpha$. As detailed in Section 1, this brute force approach can be computationally prohibitive even for moderate $N$.

## 3 Approximations

We next review various approximations to the solution of Equation (1) that are available from the literature. While many authors have considered approximating dropping a pre-defined (single) subset of data from an

---

[1]See Broderick et al. (2020) for a more general definition, including other data analyses and other quantities of interest.

analysis, we here focus on dropping the worst-case subset of data as in Equation (1); see Appendix A.3 for further discussion of this distinction and related work. We also provide a systematic comparison of theoretical running time costs; an empirical comparison of costs appears in our experiments.

### 3.1 Additive approximations

We start with what we call *additive approximations*. In particular, additive approximations (a) approximate the impact (to a quantity of interest) due to dropping a single data point and (b) add up the individual impacts to approximate the impact of dropping a group of data points.

**Approximate Maximum Influence Perturbation (AMIP).** Broderick et al. (2020) propose relaxing $w$ to allow continuous values and replacing the $w$-specific quantity of interest with a first-order Taylor series expansion with respect to $w$ around $1_N$. This approximation applies to more general data analyses and quantities of interest. In our case (cf. Appendix B.1), this approximation amounts to replacing Equation (1) with

$$\max_{w \in W_\alpha} \sum_{n=1}^{N} (1 - w_n) \frac{\partial \hat{\theta}_p(w)}{\partial w_n}\Big|_{w=1_N}. \tag{2}$$

Let $e_p$ denote the $p$th standard basis vector and $\mathbf{X} \in \mathbb{R}^{N \times P}$ denote the design matrix, where $N > P$ and we assume $\mathbf{X}$ is full rank. For OLS with an effect-size quantity of interest, $\theta_p$, the formula for the influence score of the $n$th data point is a product of a leverage-like term and a residual term,

$$\frac{\partial \hat{\theta}_p(w)}{\partial w_n}\Big|_{w=1_N} = \underbrace{e_p^\top (\mathbf{X}^\top \mathbf{X})^{-1} x_n}_{\text{leverage-like term}} \underbrace{(y_n - \hat{\theta}(1_N)^\top x_n)}_{\text{residual term}}. \tag{3}$$

For a derivation of Equation (3), see Equation (10) in Appendix B.1.

For the quantity of interest $\theta_p$ then, the AMIP approximation replaces Equation (1) with an optimization problem that can be solved by maximizing a sum of influence scores:

$$\max_{w \in W_\alpha} \sum_{n=1}^{N} (1 - w_n) e_p^\top (\mathbf{X}^\top \mathbf{X})^{-1} x_n (y_n - \hat{\theta}(1_N)^\top x_n). \tag{4}$$

The AMIP algorithm solves Equation (4) by (a) running the original data analysis, (b) computing the *influence scores* (Equation (3)), (c) finding the largest $\lfloor \alpha N \rfloor$ values, and (d) adding up the influence scores to approximate the impact of dropping the group. The approximate Most Influential Set returned here is precisely the set of points with the largest $\lfloor \alpha N \rfloor$ influence scores. The overall cost of running AMIP for a general data analysis is $O(Analysis + N \log(\alpha N) + NP^2 + P^3)$,[2] where *Analysis* represents the cost of the data analysis. The cost of running OLS is $O(NP^2 + P^3)$. So, for OLS with an effect-size quantity of interest, the cost of running AMIP is $O(NP^2 + P^3 + N \log(\alpha N))$.

**Additive One-Exact.** Kuschnig et al. (2021) approximate the change in effect size that results from dropping a group of data points in OLS by the sum of the impacts of dropping individual points; the idea may be applied more broadly, for more general losses or quantities of interest. We call this approach *Additive One-Exact*. The broad idea is to (a) compute the exact impact of dropping each single data point on the quantity of interest, (b) find the $\lfloor \alpha N \rfloor$ data points that, when dropped individually, yield the changes of largest magnitude in the desired direction, and (c) add up those individual impacts to approximate the impact of dropping the group. Here, the approximate Most Influential Set returned is the set of $\lfloor \alpha N \rfloor$ points that, when dropped individually, yield the changes of largest magnitude in the desired direction. For general losses, Additive One-Exact can cost $N$ times the cost of a single data analysis and need not be exact for $\lfloor \alpha N \rfloor > 1$. In the special case of OLS with an effect-size quantity of interest, Additive One-Exact requires just a single data analysis but still need not be exact for $\lfloor \alpha N \rfloor > 1$.

When we simultaneously consider (a) OLS linear regression and (b) the effect-size quantity of interest, we observe that Additive One-Exact can be seen as equivalent to another approximation, which we call *Additive*

---

[2]The floor function introduces discrete rounding effects that are negligible in the asymptotic regime; see Appendix B.5 for more details.

*One-step Newton* and define in Appendix B.2. Without the "Additive" descriptor, *One-step Newton* is a popular existing approximation that, in the recent machine learning literature, has been used in estimating the impact of dropping a fixed subset of data (Beirami et al., 2017; Sekhari et al., 2021; Koh et al., 2019; Ghosh et al., 2020). Additive One-step Newton adds up single-data-point approximations from One-step Newton; see Appendix B.2 for details. The special case of Additive One-step Newton for logistic regression was first proposed by Park et al. (2023) in the context of data attribution. We hope that our more general formula for Additive One-step Newton provides another angle on extending the Additive One-Exact algorithm to models beyond OLS, to the more general class of differentiable losses.

In the general data analysis setting, the computation of One-Exact scores involves re-running the data analysis upon dropping each individual point in a data set, a cost that is $O(N \times Analysis)$. In the setting of OLS, we can take advantage of the One-step Newton update in place of re-running the analysis $N$ times (see Appendix B.2 for more details). Using this rank-one update, the cost of computing One-Exact scores for $N$ data points becomes $O(NP^3 + P^3)$, or simply $O(NP^3)$. Notice the additional $P$ factor relative to the $O(NP^2)$ term in the AMIP computation; the improved precision of One-Exact scores over influence scores comes at the cost of this additional factor of $P$. Specifically, for Additive One-Exact, the Hessian matrix is reweighted to account for each dropped data point (see Equation (17) in Appendix B.2 for the equation for this approximation) while, for AMIP, this reweighting is omitted.

Thus, the general cost of running Additive One-Exact is $O(N \times Analysis + N \log(\alpha N))$, and the cost specific to OLS with an effect size quantity of interest is $O(NP^3 + N \log(\alpha N))$. See Appendix B.4 for more details.

### 3.2 Greedy approximations

Next we discuss *greedy approximations.* Greedy approximations iteratively (a) approximate the change (to the quantity of interest) upon dropping each data point individually, (b) select the point that results in the biggest approximated change when dropped, and (c) re-run the data analysis without this point (Belsley et al., 1980, Section 2.1).

**Greedy One-Exact.** The outlier detection literature has highlighted the combinatorial cost of finding influential subsets exactly. This literature also describes the *masking problem* that can arise in additive approximations of influence: namely, when one outlier hides the impact of another (Belsley et al., 1980; Atkinson, 1986). To address these issues, Belsley et al. (1980, Section 2.1) suggest to greedily remove one outlier point at a time in a stepwise procedure. Kuschnig et al. (2021) propose a similar greedy procedure for approximating the Maximum Influence Perturbation; namely, they iteratively: (a) compute the exact change (to the quantity of interest) upon dropping each data point individually, (b) select the point that results in the biggest change when dropped, and (c) re-run the data analysis (Belsley et al., 1980, Section 2.1). In general, Greedy One-Exact requires $\lfloor \alpha N \rfloor$ times the cost of an additive approximation.

For a general data analysis, Greedy One-Exact involves running $N$ re-runs of a data analysis for $\lfloor \alpha N \rfloor$ iterations. Thus, the overall cost of running Greedy One-Exact is $O(\alpha N^2 \times Analysis)$. In the OLS-specific setting, we can again take advantage of the rank-one update as described for Additive One-Exact. Specifically, to compute One-step Newton scores for $N$ data points costs $O(NP^3)$; repeated over $\lfloor \alpha N \rfloor$ iterations, step (a) costs $O(\alpha N^2 P^3)$. To find the top One-Exact score $\lfloor \alpha N \rfloor$ times, step (b) costs $O(\alpha N^2)$. One run of OLS costs $O(NP^2 + P^3)$; performed over $\lfloor \alpha N \rfloor$ iterations, step (c) costs $O(\alpha N P^3)$.

Thus, for OLS with an effect size quantity of interest, the cost reduces to $O(\alpha N^2 P^3)$. We find this cost is not prohibitive in our examples below; see Section 6 for an empirical analysis of running time. See Appendix B.4 for a more detailed description of the asymptotic running time analysis just above.

**Greedy AMIP.** We define Greedy AMIP analogously to Greedy One-Exact, replacing the exact effect of removing a point with the influence score approximation. To the best of our knowledge, Greedy AMIP has not previously been proposed in the literature, for any data analysis or quantity of interest.

For a general data analysis, running Greedy AMIP costs $\lfloor \alpha N \rfloor$ times the cost of running AMIP. This cost is $O(\alpha N \times Analysis + \alpha N^2 P^2 + \alpha N P^3)$. Substituting in the cost of OLS, we find that the cost of running Greedy AMIP for OLS with an effect size quantity of interest is $O(\alpha N^2 P^2 + \alpha N P^3)$. See Appendix B.4 for the more detailed description of these results.

### 3.3 Approximations specific to Ordinary Least Squares with effect-size quantity of interest

A line of recent works provide mathematical programs that give upper bounds on the size of the Most Influential Set for settings of OLS where the quantity of interest is an effect size (Moitra & Rohatgi, 2023; Freund & Hopkins, 2023). Unlike the additive and greedy approximations, these algorithms do not directly output the approximation of the Maximum Influence Perturbation or the Most Influential Set. Both quantities, however, are straightforward to obtain from the algorithms, as we describe next.

**NetApprox.** The NetApprox algorithm of Moitra & Rohatgi (2023) seeks to find the size of the smallest data subset that, if removed or down-weighted (this algorithm works with fractional data weights), would zero out the sign of a particular regression coefficient.

In order to use the output of NetApprox for the Maximum Influence Perturbation task, we first obtain the fractional data-weights, $w := (w_1, ..., w_N)$ where $w_n \in [0, 1]$, computed using NetApprox.[3] We set the Approximate Most Influential Set to be the set of $\lfloor \alpha N \rfloor$ points that have the smallest weights (i.e., weights closest to 0) and such that those weights are strictly less than 1.[4] We then drop the points in the approximated Most Influential Set and refit the model to find the approximated Maximum Influence Perturbation (i.e., the maximum change in the effect size that can be induced by dropping a subset of at most $\lfloor \alpha N \rfloor$ data points).

In order to make computations tractable, NetApprox works by strategically selecting a "net," a finite number of coefficient vector configurations. For every chosen configuration, the algorithm solves a linear program to determine the minimum number of samples that need to be removed in order to zero out the first regression coefficient. Running NetApprox involves solving $O(P^{P/2})$ linear programs; altogether, the runtime for this algorithm is $O(P^{P/2} \cdot \text{poly}(N))$.

**FH-Gurobi.** Freund & Hopkins (2023) provide their own implementations of the mathematical program introduced in Moitra & Rohatgi (2023). In particular, the authors implement two versions of this mathematical program: (1) a fractionally-relaxed version, where data weights can take values between 0 and 1 (inclusive), and (2) an integer-constrained version, where data weights are forced to take on the integers 0 or 1. Both versions are solved using exact solver methods supported from Gurobi 9.0 onwards (specifically, the authors note that these methods apply a globally optimal spatial branch-and-bound method that recursively partitions the feasible region into subdomains). The paper refers to these two mathematical programs as Gurobi, for the optimization software they were implemented in. For ease of distinguishing this approximation from the commercial optimization software it was implemented in, we refer to the approximation as FH-Gurobi. Freund & Hopkins (2023) found that the integer-constrained version of FH-Gurobi showed substantially worse performance for their task, so the authors recommend running it with a warm start from the rounded weights obtained using the fractionally-relaxed version.

In the experiments that follow, we compare against two versions of the algorithm Freund & Hopkins (2023) proposed for the Maximum Influence Perturbation problem. In the first version, which we call FH-Gurobi, we run the integer-constrained mathematical program and deem the approximate Most Influential Set as the set of indices with weight 0. If the size of this set is no greater than $\lfloor \alpha N \rfloor$, we refit OLS upon dropping these points and deem the difference between the original fit and the new fit to be the approximated Maximum Influence Perturbation (i.e., the maximum change in the effect size that can be induced by dropping a subset of at most $\lfloor \alpha N \rfloor$ data points). If the algorithm sets more than $\lfloor \alpha N \rfloor$ data weights to 0, then we conclude that there does not exist such a subset that can be dropped to change conclusions (i.e., the data analysis is robust). In the second version, which we call FH-Gurobi (warm-start), we first run the fractionally-relaxed mathematical program, then use those outputs (i.e., the fractional weights) as input to the integer-constrained mathematical program. We then determine the approximate Most Influential Set and Maximum Influence Perturbation in the same way as we do for the integer-constrained version.

---

[3]The original NetApprox algorithm computes, but does not return, the data weights.
[4]In all of our experiments, NetApprox returned at least $\lfloor \alpha N \rfloor$ points with weight strictly less than 1, so the question of how to handle weights equal to 1 did not arise in practice.

### 3.4 Lower bound algorithms

Just as there are algorithms that provide upper bounds on the size of the Most Influential Set, there are algorithms that provide lower bounds (Moitra & Rohatgi, 2023; Freund & Hopkins, 2023; Rubinstein & Hopkins, 2025). However, we are not able to easily determine a Most Influential Set using these algorithms, so we do not compare to these in the sections below. For more details on these algorithms, see Appendix A.6.

## 4 Failure modes

We start by defining what failure means in the present context, and then we show a range of experiments demonstrating failures for some of the approximations above.

### 4.1 What failure means here

We consider the data analyst interested in whether their analysis is robust to dropping a small fraction of data. With that in mind, we say that an approximation fails if there truly exists a small fraction of data that we can drop to change the conclusions of the analysis, but the approximation reports that such a data subset does not exist. For OLS, we expect that analysts are willing to re-run their analysis at least once (after any of the approximations defined above) with the approximate Most Influential Set dropped, so we are most concerned about the following type of failure.

**Definition 4.1.** We say there is a *failure with re-run* if (a) there exists a small fraction of data that we can drop to change conclusions and (b) we remove the points suggested by the method and re-run the analysis, but we do not see an actual change in conclusions upon re-running.

The greedy approximations (Section 3.2) and NetApprox (Section 3.3) already require the analyst to re-run their analysis with the suggested points dropped. Both FH-Gurobi algorithms also effectively require re-runs; see Appendix C.3 for a discussion of some subtleties. The additive approximations (Section 3.1), however, do not inherently require re-runs of the data analysis. Thus, the additive approximations introduce the potential for an additional type of failure, which we define next.

**Definition 4.2.** We say there is a *failure without re-run* if there exists a small fraction of data that we can drop to change conclusions, but the approximation reports that such a data subset does not exist.

We contrast these notions of failure specific to the problem of data-dropping robustness with alternative notions. For instance, Moitra & Rohatgi (2023); Freund & Hopkins (2023); Rubinstein & Hopkins (2025); Hu et al. (2024) are concerned with exact recovery of the set of data points that, if dropped, change the quantity of interest by the largest amount. However, we note that even if 2 data points out of 10,000 can be dropped to change conclusions, a practitioner might be similarly worried to hear that 3 data points can be dropped to change conclusions. Moitra & Rohatgi (2023); Freund & Hopkins (2023); Rubinstein & Hopkins (2025); Hu et al. (2024) are also concerned with larger $\alpha$ fractions, but we focus on $\alpha \leq 0.01$ since small $\alpha$ fractions are the concern of the present form of robustness (see Section 2). Finally Moitra & Rohatgi (2023) are concerned with adversarially constructed data configurations, but we focus on data configurations that could arise naturally in practice.

Our failure modes focus on *under*estimation of sensitivity; if the data analyst is willing to re-run their analysis once, any non-robustness found from that re-run is conclusive. So in this case, *over*estimation (i.e., a false positive, where the method detects non-robustness when it should not) is not a concern.

### 4.2 One outlier

We start by considering a case with a single data point far from the bulk of the data. We find experimental failures in AMIP and FH-Gurobi (without warm start) and support our findings with intuition from theory.

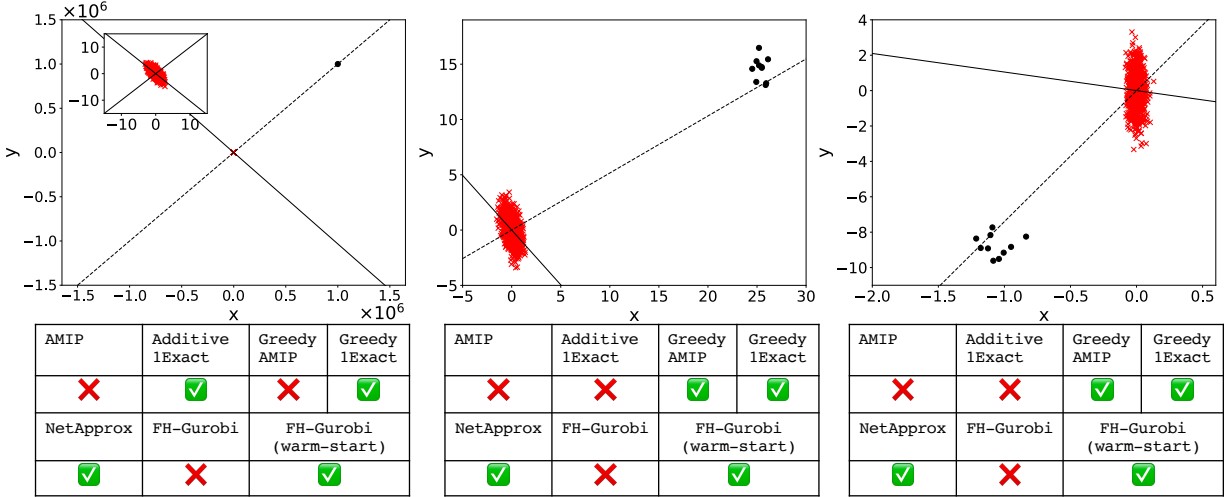

Figure 1: Our examples: one-outlier (left), Simpson's paradox (middle), poor conditioning (right). A dashed line represents the OLS-estimated slope on the entire data set while a solid line represents the slope with black dots removed. The left plot includes an inset zoomed in on the bulk of the data. The tables display the performance of the approximations for the corresponding example: a red X indicates a failure with re-run, and a green check indicates a success.

### 4.2.1  One outlier experiment

**Setup.** In realistic data settings, we may have a single data point far from the bulk of the data; this outlier may arise due to data-entry errors, machine-measurement errors, or heavy tails in both the covariate and response. To construct the plot in Figure 1 (left), we draw 1,000 red crosses by taking $x_n \sim \mathcal{N}(0, 1)$ i.i.d. and $y_n = -x_n + \epsilon_n$ with $\epsilon_n \sim \mathcal{N}(0, 1)$ i.i.d. Throughout, we use $\mathcal{N}(\mu, \sigma^2)$ to denote the normal distribution with mean $\mu$ and variance $\sigma^2$. The black dot appears at $x_n = y_n = 10^6$. We fit OLS with an intercept. The OLS-estimated slope on the full data set is nearly 1; after dropping the black point (less than 0.1% of the data set), the estimate is nearly -1, representing a sign change.

**Experimental Results.** We summarize the performance with re-run in the left table of Figure 1. When asked to find the worst-case 0.1% of the data set to drop, Additive One-Exact succeeds because, by construction, it is exact for removing a single point. AMIP chooses a red cross to drop; it predicts that no sign change will be achieved, and dropping the chosen point and re-running also does not achieve a sign change. It follows that AMIP suffers both types of failure (i.e., with and without re-run) in this example. When considering failure with re-run, there is no distinction between greedy and additive algorithms when $\lfloor \alpha N \rfloor = 1$, so Greedy One-Exact and Greedy AMIP perform the same as their additive counterparts. FH-Gurobi (without warm-start) fails while both NetApprox and FH-Gurobi (warm-start) succeed in this setting. In this experiment, failures with and without re-run align for each approximation. We provide more detail on performance in Table 1 in Appendix C.1.

The leverage of the black-dot point must be near one in order to construct this failure mode; that is, the black dot must account for most of the variance in the covariance matrix. However, we note that the exact alignment of the black dot on the line $y_n = x_n$ is incidental. In our theory next, we prove that we can expect such failures when the black-dot point is chosen with $y_n = c x_n$ for any constant $c > 0$ and sufficiently large $|x_n|$. We also note that, although Kuschnig et al. (2021) have observed that high-leverage observations can lead to problems for the AMIP, they do not demonstrate this result in a setting with one outlier or provide supporting theory.

### 4.2.2 One outlier theory

Our theory illustrates why we might expect AMIP to fail in this one-outlier case. In particular, we first demonstrate why a point far from the bulk of the data might have a low influence score. Second, we demonstrate why we might expect other data points to have higher influence scores. Together, these facts suggest why the outlier point might have a lower influence score than central points and thereby not be chosen for dropping in the approximation, potentially leading to a failure with re-run.

We do not need to assume any data-generating process to prove our results. Rather, we take a single point increasingly far away from the origin. First, observe that, in linear regression, the influence score factorizes into two terms: the residual times a leverage-like term (Hampel, 1974); see Appendix C.6 and Equation (3) for full details. A sufficiently-far outlier will have a very low residual. Meanwhile we show that the leverage-like term goes to zero as the outlier gets farther away (see Lemma C.2). So a sufficiently-far outlier will have a vanishingly low influence score. We let this outlier be the first data point in the result in Proposition 4.3 below. If this outlier caused all the other influence scores to become vanishingly low as well, it might still have the largest influence. We show this collective vanishing behavior need not happen in Example 1 below; in particular a second data point's influence score does not vanish.

Before stating our results, we introduce some notation relating to fitting OLS on all data points except the first point, which we will take to be the outlier. Let $\mathbf{X}_{-1} \in \mathbb{R}^{(N-1) \times P}$ be the design matrix with the first row deleted, and let $y_{-1} \in \mathbb{R}^{N-1}$ be the response vector with the first entry deleted. Define $\boldsymbol{A}_{-1} := \mathbf{X}_{-1}^\top \mathbf{X}_{-1}$ and $b_{-1} := y_{-1}^\top \mathbf{X}_{-1}$.

**Proposition 4.3.** *Choose any $v \in \mathbb{R}^P$ with $\|v\| = 1$ and any constant $c > 0$. Let $(x_1, y_1) = (\lambda v, \lambda c)$. Let $(x_n, y_n)_{n=2}^N$ be any points in $\mathbb{R}^P \times \mathbb{R}$ such that $\mathbf{X}_{-1}$ has rank $P$. Let $\hat{\theta}_p$ denote the pth entry of the OLS estimator, $\hat{\theta}$, fit without an intercept. Then, for all $1 \le p \le P$,*

$$\lim_{\lambda \to \infty} \left. \frac{\partial \hat{\theta}_p(w)}{\partial w_1} \right|_{w=1_N} = 0, \quad (5) \qquad and \qquad \lim_{\lambda \to \infty} \left. \frac{\partial \hat{\theta}_p(w)}{\partial w_2} \right|_{w=1_N} = \frac{st}{(v^\top \boldsymbol{A}_{-1}^{-1} v)^2}, \quad (6)$$

*where $s := (v^\top \boldsymbol{A}_{-1}^{-1} v e_p^\top \boldsymbol{A}_{-1}^{-1} x_2 - e_p^\top \boldsymbol{A}_{-1}^{-1} v v^\top \boldsymbol{A}_{-1}^{-1} x_2)$ and $t := (y_2 v^\top \boldsymbol{A}_{-1}^{-1} v - c v^\top \boldsymbol{A}_{-1}^{-1} x_2 - b_{-1} \boldsymbol{A}_{-1}^{-1} x_2 v^\top \boldsymbol{A}_{-1}^{-1} v + b_{-1} \boldsymbol{A}_{-1}^{-1} v v^\top \boldsymbol{A}_{-1}^{-1} x_2)$.*

Equation (5) is the limiting value of the influence score for the first data point (the outlier). And so Proposition 4.3 tells us that under mild conditions, an extreme outlier has a small influence score. Equation (6) is the limiting value of influence score of an (arbitrary) other point in the data set, and we see that it converges. When its limit is not equal to 0, Proposition 4.3 implies that, for extreme enough outliers in the response and covariate directions, the influence score of the outlier point will be smaller in magnitude than that of another point in the data set. Note, when $P = 1$, $s$ is always equal to 0, so the right hand side of Equation (6) is also 0. Next, we give an example (in $P > 1$) where $s, t \ne 0$, and so the righthand side of Equation (6) is nonzero.

**Example 1.** Consider the data set with $\mathbf{X} = \begin{bmatrix} \lambda & 0 \\ 3 & 4 \\ 5 & 6 \end{bmatrix}$, $y = \begin{bmatrix} \lambda \\ 2 \\ 3 \end{bmatrix}$. Suppose we are making a decision based on the sign of the second effect, $p = 2$. In this setting,

$$\lim_{\lambda \to \infty} \left. \frac{\partial \hat{\theta}_p(w)}{\partial w_2} \right|_{w=1_N} = \frac{st}{(v^\top \boldsymbol{A}_{-1}^{-1} v)^2} = 0.0178 > 0. \quad (7)$$

*Proof.* We have $v = \begin{bmatrix} 1 \\ 0 \end{bmatrix}$, $c = 1$, $x_2 = \begin{bmatrix} 3 \\ 4 \end{bmatrix}$, $y_2 = 2$. It follows that $\boldsymbol{A}_{-1} = \begin{bmatrix} 34 & 42 \\ 42 & 52 \end{bmatrix}$, $s = 1$, $t = 3$, and $(v^\top \boldsymbol{A}_{-1}^{-1} v)^2 = 169$. $\qquad \square$

While these theoretical results are presented without an intercept and our numerical results are fit with an intercept, we find similar failure modes regardless of the inclusion of the intercept. See Appendix C.2 for a version of the numerical results fit without an intercept and a more nuanced discussion on the impact of the intercept term.

### 4.3 Multiple outliers

We next identify two realistic cases with multiple outliers far from the bulk of the data such that the additive approximations and FH-Gurobi (without warm start) fail. We support our empirical findings with theory. Essentially we see how failure changes if we have a small group of outliers instead of just a single outlier.

#### 4.3.1 Simpson's paradox

It is common to have (at least) two noisy subpopulations within a single data set; we consider the case where one subpopulation represents a small fraction of the total. For instance, we might have heterogeneity in the population that the regression model does not account for; Simpson's paradox describes the case when the trend within subpopulations reverses the trend across the full populations.

**Setup.** In the particular example in Figure 1 (middle), the overall slope (across all the data) has a different sign than the slope in just the red data or just the black data. To create the illustration in Figure 1 (middle), we draw 1,000 red crosses with $x_n \sim \mathcal{N}(0, 0.25)$ i.i.d., $y_n = -x_n + \epsilon_n$, and $\epsilon_n \sim \mathcal{N}(0, 1)$ i.i.d. We draw 10 black dots with $x_n \sim \mathcal{N}(25, 0.25)$, $y_n = -x_n + 40 + \epsilon_n$ i.i.d., and $\epsilon_n$ as before. The OLS-estimated slope on the full data set is 0.52. Dropping the black dots (1% of the data) yields a slope of -0.99, a sign change.

**Experimental results.** We summarize the performance with re-run in the middle table of Figure 1. When asked to find the worst-case 1% (i.e., 10 data points) of the data set to drop, both AMIP and Additive One-Exact choose some red-cross points and some black-dot points to drop; see Table 2 in Appendix C.1 for full detail. Both methods predict there will be no sign change (a failure without re-run). Upon removing the flagged data points and re-running the data analysis, in both cases we find no sign change (a failure with re-run). At extra computational expense, both greedy methods flag exactly the black-dot data points as the points to drop, so neither suffers a failure. At the cost of further increasing compute time, NetApprox and FH-Gurobi (warm-start) both report that a data subset of size 10 exists to flip the sign, while FH-Gurobi without warm-start fails to report the existence of such a subset.

**Discussion.** Once we leave the regime of one data point, we see that both additive methods (AMIP, Additive One-Exact) and FH-Gurobi (without warm start) can fail. We see that errors can arise when we approximate the change in dropping a group of data points by the sum of the changes of dropping individual data points. This phenomenon is known more broadly as masking, where one outlier can hide the effect of another (Belsley et al., 1980; Atkinson, 1986). To overcome masking problems, previous work has noted the success of using greedy procedures, both in problems of outlier detection (Hadi & Simonoff, 1993; Lawrance, 1995) as well as in the problem of identifying the Maximum Influence Perturbation (Kuschnig et al., 2021). Although both of our multi-outlier examples (Simpson's paradox here and poor conditioning below) demonstrate the phenomenon of masking, the failure modes we surface are distinct from the simulation studies of Kuschnig et al. (2021); our examples demonstrate settings where removing a small fraction of the data can lead to a change in sign of the regression coefficient—a failure of the approximation, according to our definition in Section 4. The examples in Kuschnig et al. (2021) demonstrate cases where AMIP can misestimate the change of an effect's magnitude in the face of masking. But those examples demonstrate neither a failure without re-run or a failure with re-run. See Appendix A.5 for more discussion on masking.

#### 4.3.2 Poor conditioning

We do not expect that the alignment of data points within the outlier cluster (black dots) in the example above will be germane to our results. However, recall that the OLS objective is not rotationally invariant in the two-dimensional space defined by a single covariate and response. So it is a priori possible for the alignment of the relative trend in the bulk of the data (red crosses) and the trend across the full data to matter. In particular, we next consider a case where the bulk of the data is ill-conditioned on its own.

Poorly-conditioned data are a common concern to users of OLS regression (Chatterjee & Hadi, 1986). Here, we adapt an example presented in Moitra & Rohatgi (2023). In the example of Moitra & Rohatgi (2023), the data points were constructed to lie perfectly along two straight lines (see Figure 7). Moreover, the small subset of outlier data points lie perfectly along the OLS-estimated slope for the entire data set. The removal

of the black points causes all variation along covariate space to be lost and the OLS solution to become ill-defined.

**Setup.** We alter the adversarial setup of Moitra & Rohatgi (2023) into one that might arise in natural data settings with no adversary (see Figure 1 (right)). To that end, we add generous amounts of noise to both red-cross points and black-dot points and translate the black-dot points to no longer lie along the OLS-estimated slope. We generate the red crosses so as to have poor conditioning; since there is much more noise around the (zero) trend than variation in the covariates, there is no clear regression solution. In particular, we generate the 1,000 red crosses with $x_n \sim \mathcal{N}(0, 0.001)$ i.i.d., $y_n = \epsilon_n$, and $\epsilon_n \sim \mathcal{N}(0, 1)$ i.i.d. We draw the 10 black dots as $x_n \sim \mathcal{N}(-1, 0.01)$ i.i.d., $y_n = -x_n - 10 - \epsilon_n$, and $\epsilon_n$ as before. When we consider both black dots and red crosses together as a single data set, there is no poor conditioning. The OLS-estimated slope on the full data set is around 7.40; dropping the black dots (1% of the data) yields a slope of about -1.04, a sign change.

**Experimental results.** We summarize the performance with re-run in the right table of Figure 1. We ask each method whether it is possible to drop at most 1% (10 data points) of the data set and change the sign of the effect. Both AMIP and Additive One-Exact choose some red crosses and some black dots; see Table 3. Both methods in turn suffer failures with and without re-run. The greedy methods are more computationally expensive but succeed. NetApprox and FH-Gurobi (warm-start) both report that a data subset of size 10 exists to flip the sign, while FH-Gurobi without warm-start fails to report the existence of such a subset.

### 4.3.3 Theory

A common theme in our multi-outlier examples is that, in general, the impacts of data-dropping are non-additive. Our theory illustrates that, even in a simple setup, data points can mask each other's impacts. In particular, our theory shows we can have masking issues even in OLS with just a single covariate and no intercept, and even between just two outlier data points. The issues in our one-outlier theory (Proposition 4.3) could be overcome by using a One-Exact method. But our next results show that Additive One-Exact falls prey to masking issues.

**Proposition 4.4.** *Let $\lambda, c \in \mathbb{R}$. Consider a pair of data points, $(x_1, y_1) = (\lambda, \lambda)$ and $(x_2, y_2) = (\lambda, \lambda + c)$. Let $(x_n, y_n)_{n=3}^N$ be any points in $\mathbb{R} \times \mathbb{R}$ such that at least one of $(x_n)_{n=3}^N$ is non-zero. We apply OLS to the single covariate $x$ and response $y$ with no intercept; we make a decision based on the sign of the resulting effect size. As $\lambda \to \infty$, the Additive One-Exact approximation (Section 3.1) to the change in effect size from dropping $(x_1, y_1), (x_2, y_2)$ tends to zero, while the true change in effect size tends to $1 - \left( \sum_{n \neq 1,2}^N x_n y_n / \sum_{n \neq 1,2}^N x_n^2 \right)$.*

In the setup of Proposition 4.4, $P = 1$, and there is no intercept. So the assumption that at least one of $(x_n)_{n=3}^N$ is non-zero is equivalent to an assumption that the design matrix with the first two points removed is full rank. That is, the assumption ensures the OLS solution remains well-defined after dropping the first two data points. See Appendix C.7 for the proof of Proposition 4.4.

Proposition 4.4 tells us that, as a pair of points is taken to infinity together, the sum of their individual impacts (i.e., the change in effect size from dropping each point individually) always approaches zero, regardless of what the group impact (i.e., the change in effect size from dropping the pair together) approaches. Indeed, the impact of dropping the group may result in a substantive change in effect size, information that cannot be gleaned solely from looking at individual impacts.

While the theory in this section is presented without an intercept term, we find that the exclusion of the intercept has very little effect on the numerical results presented in Section 4. In particular, we run versions of the numerical results fit without an intercept term in Appendix C.2 and find little difference.

## 4.4 Greedy failures

Moitra & Rohatgi (2023, Section 5.1) construct a failure mode of greedy approximations in an adversarial way. We find it helpful to visualize their (text) construction; see Figure 7 in Appendix C.7.2. We surface similar adversarial examples where greedy algorithms fail; see Figure 8 and Figure 9. But we have yet to

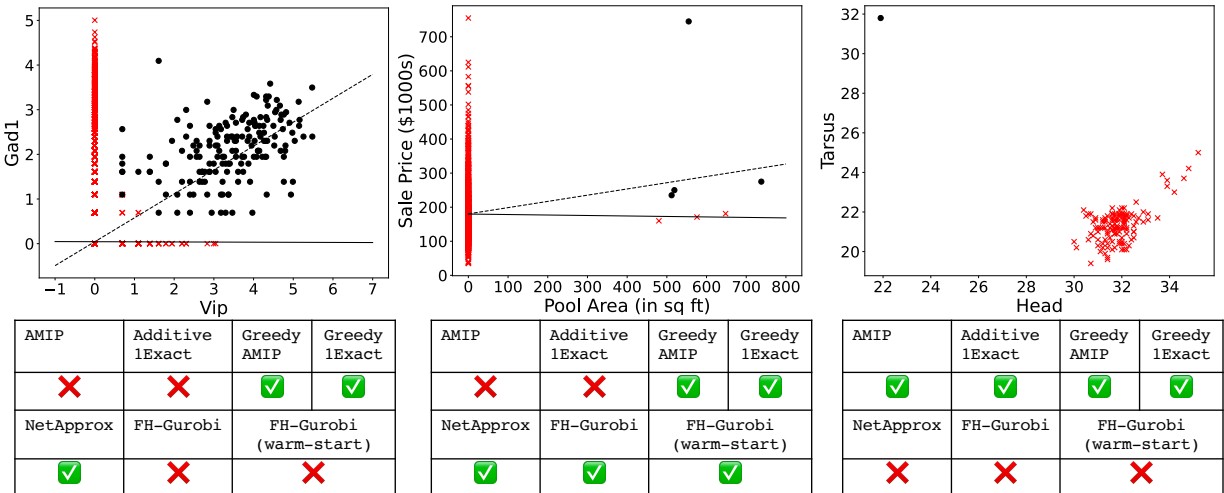

Figure 2: Our examples: Single-cell Genomics (left), Ames Housing (middle), Bird Morphometrics (right). A dashed line represents the OLS-estimated slope on the entire data set while a solid line represents the slope with black dots removed. We do not plot fitted models for the Bird Morphometrics data since the regression involved two additional covariate dimensions. The tables display the performance of all approximations for the corresponding example: a red X indicates a failure with re-run, and a green check indicates a success.

identify realistic, non-adversarial examples of greedy failures beyond the dropping of a single data point. Furthermore, we were unable to identify *any* realistic failures of Greedy One-Exact.

## 5 Examples in real-world data

We next illustrate failures in real-data analyses. We were able to surface failures in real-data analyses for all but the greedy approximations. We also illustrate various real-data analyses where all methods succeed in Appendix C.4. In each of the following real-world examples, we know that we can drop less than 1% of the data and change the sign of the OLS regression coefficient since at least one method successfully identifies such a set. We check whether all methods do so.

The Single-cell Genomics and Ames Housing data sets demonstrate multi-outlier failure modes, while the Bird Morphometrics example demonstrates a one-outlier failure mode.

### 5.1 Single-cell Genomics

**Setup.** Our first data set is taken from a study on the impact of sensory experiences on gene expression in the mouse visual cortex (Hrvatin et al., 2018). The data set contains a total of 65,539 points. As is common in gene expression data, these data are heavily zero-inflated; 99.5% of the Vip gene values (shown on the x-axis in Figure 2 (left)) are 0, and 97.39% of the Gad1 gene values (shown on the y-axis) are 0. Practitioners are often interested in the association between two genes. We consider a linear regression (with intercept) of Gad1 values on Vip values.

This data analysis is not robust to dropping 1% of the data. In particular, there exists a subset of size 172 (0.26% of the data) that, when dropped, can change the sign of the regression coefficient from positive (0.536) to negative ($-0.003$). We plot these 172 points as black dots in Figure 2 (left). We plot the remaining points of the data set as red crosses. Note that, due to zero-inflation, many data points are stacked at $(0, 0)$.

**Experimental results.** We summarize the performance with re-run in the left table of Figure 2. We ask each method whether it is possible to drop at most 1% (656 data points) of the data set and change the sign of the effect. Both AMIP and Additive One-Exact predict there will be no sign change (a failure without

re-run). Upon removing the flagged worst-case 1% of data points and re-running the data analysis, in both cases we still find no sign change (a failure with re-run). Both of the greedy methods, however, successfully identify 172 data points to drop to change the sign, so neither suffers a failure. NetApprox also successfully identifies such a set. FH-Gurobi returns a set of size 1712 to drop, which is larger than 1% of the data, so it fails here. Interestingly, FH-Gurobi (warm-start) does successfully identify a set of size 172 points to drop, but the subset it returns is incorrect in that, when dropped, the regression coefficient is still greater than 0 (it was 0.00150), and thus no sign change is detected. See Table 4 in Appendix C.1 for full details.

## 5.2 Ames Housing

**Setup.** The Ames Housing data set provides a comprehensive collection of residential property data from Ames, Iowa, and is a widely utilized data set for regression modeling exercises (De Cock, 2011). The response variable here is SalePrice, the final selling price of each home, while the covariates consist of building, land, and facility characteristics. We perform a one-dimensional linear regression (with intercept) of SalePrice on the pool area covariate (i.e., the pool area of each property in square feet). The training data set on Kaggle contained 1,460 points.

This data analysis is not robust to dropping 1% of the data. Dropping 4 out of the 1,460 (0.27%) points changes the sign of the regression coefficient from positive (182.71) to negative ($-15.09$). In Figure 2 (middle), we plot the four points as black dots, and the rest of the data set we plot as red crosses.

**Experimental results.** We summarize the performance with re-run in the middle table of Figure 2. When asked if it is possible to drop at most 1% (14 data points) of the data and change the sign, both AMIP and Additive One-Exact predict that there will be no sign change (a failure without re-run). Upon removing the flagged worst-case 1% of points and re-running OLS, in both cases we still find no sign change (a failure with re-run). Both of the greedy methods, however, successfully identify 4 data points to drop to change the sign, so neither of these greedy methods fails. All of the OLS-specific methods succeed. See Table 5 in Appendix C.1 for full details.

## 5.3 Bird Morphometrics

**Setup.** This data set is taken from an ecological study on the morphometric features of the saltmarsh sparrow *Ammodramus caudacutus* (Zuur et al., 2010). Ecologists commonly use linear models to understand the association between animal features. The outlier in this data set may be a different species, a typing mistake, or indeed a correct record (Zuur et al., 2010).

The original data set contains 1,295 points. In order for 1 data point to account for roughly 1% of the total, we sampled 10% of the full data (129 points) uniformly at random without replacement; it happened that our first random sample retained the outlier point, so we kept this single sample. Then, we ran OLS regression (with intercept) on the features "head" (head length), "wingcrd" (wing length), and "culmen" (beak length). Removing just the single black-dot point (0.77% of the data) in Figure 2 (right) is sufficient to change the sign of the regression coefficient for "head" from negative ($-0.69$) to positive (0.399). We plot the remainder of the data set we consider as red crosses.

**Experimental results.** We summarize the performance with re-run in the right table of Figure 2. When asked to find the worst-case 1% of the data set to drop, both AMIP and Additive One-Exact succeed, as do both greedy approximations. NetApprox, however, does not identify the point that, when dropped, changes

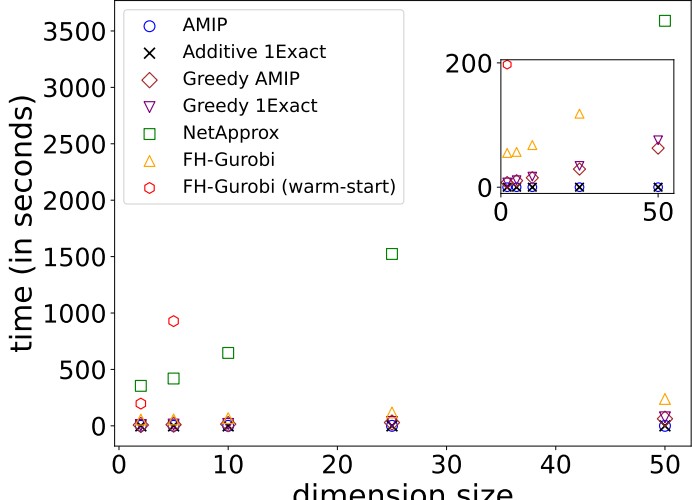

Figure 3: Plot of approximation runtimes on a simulated data set of size $N = 75,000$. We omit results that take over one hour to run; in particular, FH-Gurobi (warm-start) takes over one hour on dimension sizes 10 or larger. The plot includes an inset zoomed in on algorithms that run in under 200 seconds (i.e., the greedy and additive algorithms, as well as particular settings of FH-Gurobi without warm start).

the sign. Similarly, both versions of FH-Gurobi fail here, with each choosing a different subset of 4 points to drop. Since the subset identified is $> 1\%$ of the data, both constitute a failure.[5]

# 6 Runtime comparison

We next compare running times across approximations. In addition to being the sole algorithm that did not fail any of our tests, we find that Greedy One-Exact can be orders of magnitude faster than the OLS-specific approximations. All experiments were conducted in Python 3 on a personal computer equipped with an Apple M1 Pro CPU at 3200 MHz and 16 GB of RAM.

**Setup.** We simulate data sets of size $N = 75,000$, and we vary dimension $P$ up to $P = 50$. We choose both $N$ and $P$ to be larger than those of any data set considered by Broderick et al. (2020) and any data set in the present paper. Let $\mathbf{I}_P$ denote the identity matrix of dimension $P$ and $\mathbf{1}_P$ denote the the one-vector of dimension $P$. We draw samples with $x_n \sim \mathcal{N}(0, \mathbf{I}_P)$ i.i.d., $y_n = \langle x_n, \mathbf{1}_P \rangle + \epsilon_n$, and $\epsilon_n \sim \mathcal{N}(0, \mathbf{I}_P)$ i.i.d.

**Results.** We show the runtimes of different approximations in Figure 3. The additive algorithms (AMIP and Additive One-Exact) run the fastest for all dimensions. At dimension $P = 50$, AMIP runs for 0.03 seconds and Additive One-Exact for 0.05 seconds. The greedy algorithms are the next fastest. At dimension $P = 50$, Greedy AMIP runs for just over a minute (62.84 seconds) and Greedy One-Exact for 75.54 seconds. The third fastest is FH-Gurobi without warm-start. At $P = 50$, this algorithm runs for 238.20 seconds, or just under 4 minutes. NetApprox and FH-Gurobi (warm-start) are substantially slower than the other algorithms. At $P = 50$, NetApprox runs for 3589.93 seconds (just over 59 minutes). At $P = 10$, the FH-Gurobi (warm-start) algorithm is unable to run in under 1 hour; it ran for 3968.39 seconds, or just over 66 minutes. We are unable to run FH-Gurobi (warm-start) for dimensions much larger than $P = 10$ in our time budget. Notably, in the present experiment, Greedy One-Exact can be over 47 times faster (at $P = 50$) than the OLS-specific methods.

---

[5]When running a 1D version of OLS (fit with an intercept) on the bird morphometrics data set with head length as the sole predictor, all methods are able to succeed.

## 7   Discussion

In the present work, we identify non-adversarial failure modes of approximations to the Maximum Influence Perturbation. We focus on linear regression fit with ordinary least squares and where the decision is made based on the sign of an effect. For users interested in the Maximum Influence Perturbation for this case, we recommend the following: (1) running Greedy One-Exact if the user is willing to incur the computational expense and (2) that users visualize their data with diagnostic plots (e.g., scatter plots, leverage plots, residual plots). Our recommendation of Greedy One-Exact agrees with that of Kuschnig et al. (2021), though the notion of failure guiding our comparison is different.

Across our experiments, we find that Greedy One-Exact is able to successfully detect non-robustness when faced with both synthetic and real-world data examples. Additionally, we find that it is orders of magnitude faster than the OLS-specific approximations on plausibly-sized data sets. Our experiments suggest that, for data sets with $N \leq 75,000$ and $P \leq 50$, Greedy One-Exact should not take much longer than a minute to run; we believe this running time cost should not be prohibitive for users. Moreover, Greedy One-Exact is conceptually straightforward and should be straightforward to implement in practice; unlike NetApprox and FH-Gurobi, it does not require the use of commercial software that may not be accessible to all users.

Since NetApprox and FH-Gurobi were originally designed to estimate upper bounds on the *number* of points that must be removed to induce a sign change in a regression coefficient, it is not surprising that these algorithms might struggle with identifying a particular small subset of data points to drop to achieve a sign change. For example, in the bird morphometrics dataset, NetApprox accurately estimates the number of points that need to be dropped but fails to pinpoint the specific data point whose removal flips the sign. A similar issue arises with FH-Gurobi (warm-start) on the single-cell genomics data. Additional failure cases for both FH-Gurobi variants can be attributed to the looseness of the returned upper bounds.

While the additive methods also fail our tests, we note that AMIP can be over 2500 times faster than Greedy One-Exact in our experiments here with $N = 75,000, P = 50$. Users with larger problems (in $N$ or $P$) and smaller compute-time budgets may still find it useful to run additive approximations if the Greedy One-Exact becomes prohibitive in cost. We find that the additive methods (AMIP, Additive One-Exact) tend to fail in the presence of points with extreme leverage scores, as seen in the scatter plots in Figure 1 and in the residual-leverage plots in Figure 6. While AMIP may not be able to detect such non-robustness, a simple visualization of the data certainly would. This finding further highlights the importance of visualizing the data alongside running data-dropping robustness checks.

Not only is linear regression widely used in practice, but we believe that finding approximation failure modes in such a simple and intuitive analysis should lead us to suspect failure modes in more complex analyses until proven otherwise. It remains to investigate how alternative models (beyond low-dimensional linear regression) might interact with different data arrangements to affect approximation quality. Geometries of interest include those arising from high-dimensional covariates, generalized linear models (and other cases with constrained residuals), constrained parameter spaces (e.g., for variance parameters),[6] mixed-effect models (related to Bayesian hierarchies),[7] and other more-complex models. Across our synthetic and real-world data sets, we find that the leverage scores of the black-dot points are extremely large; see Figure 6. High-leverage points may not have the same impact on an analysis under alternative geometries. Moreover, it remains to investigate approximation performance for decisions beyond the sign of an OLS-learned effect; for instance, decisions based on statistical significance or Bayesian posterior means and variances are especially widespread. Finally, it remains to investigate the speed and accuracy trade-offs in additive and greedy approaches for data analyses more computationally expensive than the low-dimensional linear regression examples considered here; there exist many cases where a practitioner is not willing to re-run their data analysis $\lfloor \alpha N \rfloor$ times.

---

[6]This geometry was flagged as challenging in Broderick et al. (2020, Section 4.4.2).
[7]This geometry was flagged as potentially challenging in Nguyen et al. (2024, Section 6.3).

## Acknowledgments

This work was supported in part by an ONR Early Career Grant, the MIT-IBM Watson AI Lab, the NSF TRIPODS program (award DMS-2022448), and a MachineLearningApplications@CSAIL Seed Award. We are grateful to Tom Rainforth, Vishwak Srinivasan, Ittai Rubinstein, and Ryan Giordano for helpful discussions.

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

## A Related works

### A.1 Related notions of robustness

In this section, we discuss related notions of robustness and explain why robustness to worst-case data dropping provides a new and useful check on generalizability. Many tools in statistics, such as p-values and confidence intervals, are meant to measure the generalizability of sample-based conclusions (Fisher, 1925). Similarly, works on algorithmic stability in the learning theory literature quantify an algorithm's generalization error (Bousquet & Elisseeff, 2002). However, these tools rely on an assumption that the data are drawn independently and identically (i.i.d.) from the underlying target population. In most real world settings, we cannot assume the population that the samples are drawn from is identical to the target population. For instance, we might look to apply the conclusions from a specific sample to a slightly different future population. Departing from the i.i.d. regime, we can no longer rely on the theory behind classical tools alone to tell us something definitive about the generalizability of sample-based conclusions. Prior works have also studied the robustness of estimators to gross outliers and adversarially corrupted samples, arbitrary corruptions of a data-point or small collection of data points (Hampel, 1974; Madry et al., 2018; Liu & Moitra, 2022). The influence function has played a central role in the study of gross outlier sensitivity since the pioneering work of Hampel (1974). Specific to linear regression, Cook (1977) introduced Cook's Distance, for detecting outliers and gross errors. However, conclusions may still fail to generalize in the absence of gross outliers (Broderick et al., 2020). As these related notions of robustness alone do not provide a comprehensive check, a data analyst might still worry about generalizability if dropping a very small fraction of the sample can lead to drastically different conclusions.

### A.2 Computational difficulties of determining robustness to worst-case data dropping

An exact computation of the Maximum Influence Perturbation is computationally intractable. A brute force approach involves enumerating over every possible data subset, which amounts to rerunning $\binom{N}{\lfloor \alpha N \rfloor}$ data analyses, where $N$ is the number of points in the dataset. Inspired by the Maximum Influence Perturbation problem, Moitra & Rohatgi (2023); Freund & Hopkins (2023); Rubinstein & Hopkins (2025) tackle a slightly different but related problem, one of finding the minimum number of samples (in a fractional sense) that need to be removed to zero out a particular regression coefficient. For OLS regression, Moitra & Rohatgi (2023, Theorem 1.3) shows that there is no $N^{o(P)}$ time algorithm that, given a non-negative integer $k$, can determine whether the *minimum* number of samples that need to be dropped to zero out a regression coefficient is less than or equal to $k$.[8] More specifically, Moitra & Rohatgi (2023) shows that this problem requires $N^{\Omega(P)}$ time. Although their complexity result applies to the slightly different problem of determining the *minimum* number that need to be dropped to change the sign of the regression coefficient, this hardness result also applies to our problem of determining the existence of a subset of size at most $100\alpha\%$ of the data that can be dropped to change the sign. Consider the following reduction. Suppose there existed an algorithm that solves the existence problem in faster than $N^{\Omega(P)}$ computations. Specifically, for some $k$ less than $100\alpha\%$ of the data, the algorithm determines the existence of a subset of size at most $k$ that can be dropped to change the sign of the regression coefficient. If such an algorithm were to exist, we could run it for $k \in [1, \lfloor \alpha N \rfloor]$ (an operation that is $O(N)$) to determine the minimum number of samples required to be dropped to zero out the regression coefficient. The existence of such an algorithm would contradict Moitra & Rohatgi (2023, Theorem 1.3), which shows that such a task requires at least $N^{\Omega(P)}$ computations. Thus, determining robustness to

---

[8]Moitra & Rohatgi (2023) worked with a fractionally relaxed version of this problem, where the weight of a data point can take on non-integer values. This result precludes the existence of a faster solution in the integer version, as one could use the integer version to solve the weighted version (up to an approximation) by making several copies of the dataset.

small-fraction data-dropping is expensive, even in the simple setting of OLS linear regression. This prompts the need for approximations.

### A.3 Approximations to (non-worst-case) data dropping

In this section, we discuss tangential works that use approximations to data dropping in settings where the subset to drop is known. Since we are concerned with developing algorithms to overcome the combinatorial problem of searching for some worst-case subset to induce the largest change in a quantity of interest, the works discussed in this section do not provide a fast way to search for the worst-case data subset to drop.

The idea of using approximations to data dropping goes as far back as Jaeckel (1972) and Hampel (1974), who introduced the influence function in the context of robust statistics. Shortly after, Cook (1977) introduced influence measures, such as Cook's Distance, in the context of detecting outliers and gross errors. Pregibon (1981) introduced the One-step Newton approximation in the context of logistic regression diagnostics.

As models become increasingly expensive to run and datasets increasingly large, data-dropping approximations have gained popularity in several areas of machine learning. Many works have developed gradient-based approximations for cross validation (Beirami et al., 2017; Rad & Maleki, 2020; Giordano et al., 2019; Stephenson & Broderick, 2020; Ghosh et al., 2020). Works in the data privacy space have used approximations for deleting user data from models (Guo et al., 2020; Sekhari et al., 2021; Suriyakumar & Wilson, 2022). Within the model interpretability and data attribution space, methods such as Shapley value estimators (Ghorbani & Zou, 2019) and datamodels (Ilyas et al., 2022) require retraining the model a large number of times on different subsets of the data in order to quantify the impact of particular training points on the model output. In response, several works have proposed using approximations to retraining based on the influence function (Koh & Liang, 2017; Koh et al., 2019; Park et al., 2023). These gradient-based based approximations achieve great gains in computation while maintaining comparable accuracy to methods that rely on model retraining, as shown in Park et al. (2023, Figure 1). While these works (Koh & Liang, 2017; Koh et al., 2019; Park et al., 2023) investigate the performance of data dropping approximations for the task of dropping a pre-defined subset, we investigate the performance of data-dropping approximations for the task of dropping a *worst-case* data subset. The performance of an approximation may be quite different on an average-case compared to the worst-case data subset.

### A.4 Case Influence Analysis

The case influence analysis literature studies the importance of individual or groups of cases (data points) on posterior distributions (Bradlow & Zaslavsky, 1997; Carlin & Polson, 1991; Zhu et al., 2012), predictive distributions (Johnson & Geisser, 1983), and likelihoods (Cook, 1986; Zhu et al., 2007). Like the recent works on Maximum Influence Perturbation (MIP) (Broderick et al., 2020; Kuschnig et al., 2021), these works use first-order approximations to avoid re-running a full model; while the broad aim of both lines of work is to assess the impact of deleting cases on inferential results in a data analysis, there are a handful of notable distinctions between these works.

Works that develop approximations to the MIP propose algorithms to solve the optimization problem of searching for the worst-case data subset, i.e. the subset of data whose removal maximally changes some user-specified inferential quantity. Works in the case influence literature do not present solutions to this worst-case data-subset search. Rather, the case influence literature focuses on the impact of deleting observations that are specified in advance. While Zhu et al. (2012, Theorem 2) uses a first-order approximation that appears to asymptotically treat the case of dropping a non-vanishing proportion of the data, Theorem 2 is proved for a particular dropped subset, rather than a uniform bound over all subsets of a particular size, a key property allowing AMIP to approximate the deletion of the worst-case data subset. This is again shown in simulation studies, where Zhu et al. (2012) work with pre-specified subsets, namely those observations within a group in a Bayesian hierarchical model, and compares the posterior summary statistics obtained after deleting these subsets to those on the full data.

Furthermore, works in case influence typically concern whole-model sensitivity metrics—the likelihood function (Cook, 1986; Zhu et al., 2007), posterior distribution (Zhu et al., 2012), and predictive distribution (Johnson & Geisser, 1983), rather than a particular inferential quantity, such as a particular OLS regression coefficient. Sensitivity of a single regression coefficient is not sufficient to imply sensitivity to more global objects, like the entire vector of coefficients. Works on approximations to the MIP instead let the analyst specify any inferential target—often a single quantity that drives a data analysis conclusion, such as the sign or significance of one OLS coefficient—and asks how fragile that quantity of interest is to worst-case data deletion.

## A.5 Masking

We identify cases where masking, a situation where one outlier "hides" the effect of another outlier, can interfere with finding the Most Influential Set. Masking itself has been a widely studied phenomenon in the context of multi-outlier detection. Hampel (1974), a foundational text in robust statistics, notes that masking can make the identification of multiple outliers cumbersome and erroneous (Hampel, 1974, Section 1.4). Belsley et al. (1980, Section 2.1) also discussed the masking phenomenon and proposed a stepwise procedure for identifying groups of influential outliers. Bendre (1989) noted that masking can change the results for some common multiple outlier tests. Lawrance (1995) noted that masking can create errors in common diagnostic quantities, such as Cook's Distance. In the context of outlier detection, Atkinson (1986) proposed a solution that is able to mitigate the effects of masking: it is based on a two-step procedure that first fits subsamples of the data using least median of squares regression, which identifies potential groups of outliers, then uses single-point influence measures to confirm whether the points identified are indeed outliers. Gray & Ling (1984) note that the off-diagonal entries of the Hat matrix contain information about pairwise interaction effects between data points and propose an algorithm that uses this information in an attempt to overcome masking when identifying influential subsets. Both Lawrance (1995); Chatterjee & Hadi (1988) examine the the deletion of a pair of cases. Lawrance (1995) focused solely on Cook's distance as the quantity of interest, while Chatterjee & Hadi (1988) examines a range of influence measure. The works find that the degree of masking between two data points is a function of the residuals, the leverages, and the off-diagonal entries of the Hat matrix. To address masking effects in this work, we consider a stepwise approach (Greedy AMIP and One-Exact) similar to the one taken in Belsley et al. (1980, Section 2.1) to overcome a combinatorial problem of searching for a Most Influential Set. The concurrent work of Hu et al. (2024) present theory on the non-additivity of data-dropping as well, but their focus is on most influential subset selection as opposed to detecting sensitivity of statistical conclusions to worst-case small-fraction data dropping. Finally, Kuschnig et al. (2021) compare greedy and additive approximations in context of worst-case small-fraction data dropping; we build on their work by (1) identifying instances where these methods result in conclusive failures through the definitions in Section 4, and (2) providing mathematical insight into these failure modes, as we present in Section 4.

## A.6 Lower bound algorithms

Following Broderick et al. (2020), a line of works (Moitra & Rohatgi, 2023; Freund & Hopkins, 2023; Rubinstein & Hopkins, 2025) provide lower bounds on the number of points that must be removed to zero out a particular regression coefficient in OLS linear regression. However, these lower bound algorithms (1) do not identify a Most Influential Set and (2) do not compute an approximation to the Maximum Influence Perturbation, so we do not compare to these methods in this work.

However, the line of algorithms providing lower bounds can inspire future methodological development for the Most Influential Set problem in OLS. For example, Freund & Hopkins (2023) introduce a spectral algorithm that takes a more global approach than those taken by either the additive or greedy approximations. As another example, Rubinstein & Hopkins (2025) introduce a lower bound algorithm based on analyzing the error term between the AMIP approximation and the true effect of rerunning an analysis without the dropped subset. They then provide new ways to upper bound the error term expression, offering mathematical insight into the error accrued by using influence-functions based approximations to the Maximum Influence Perturbation. Despite these connections, there is currently no direct way to use these algorithms to identify a Most Influential Set, so it is not the focus of this paper.

### A.7  Failure modes

Past works have pointed out cases where worst-case data-dropping approximations may perform poorly, but these works do not define notions of failure within the context of generalization of sample-based conclusions. We define these notions of failure concretely in Section 4 and then surface examples of failures with respect to these specific definitions. Whereas previous works have pointed out failures in adversarial examples and in settings of dropping larger data fractions $> 1\%$ (Moitra & Rohatgi, 2023; Freund & Hopkins, 2023), we are concerned with failures that might arise in natural data settings without an adversary and (for purposes of generalization) where dropping a surprisingly small fraction of data leads to changes in conclusions. Certain past works have surfaced failure modes of AMIP in real-world settings without further investigations. Specifically, Broderick et al. (2020, Section 4.3) and Nguyen et al. (2024, Section 6.2) point to settings where the approximation performs poorly on a study on microcredit (Angelucci & De Giorgi, 2009). Broderick et al. (2020, Section 4.3) points out a failure of the AMIP in a setting where the quantity of interest is a hypervariance parameter in a hierarchical model. Here, AMIP approximates a positive effect while the actual refit gives a negative effect. This failure mode has another layer of distinction from our problem, as it points to a failure that may arise due to a constrained parameter space. Nguyen et al. (2024, Section 6.2) point out a setting where the approximations perform poorly for a component of a hierarchical model fitted with MCMC; in particular, they identify a setting where the confidence interval for AMIP undercovers. Finally, for the same microcredit study, Kuschnig et al. (2021); Moitra & Rohatgi (2023); Freund & Hopkins (2023) compare the performance of different approximation algorithms but not within the context of the failure definitions laid out in Section 4.

## B  Approximation supplementals

### B.1  AMIP supplementals

Broderick et al. (2020) consider a linear approximation to dropping data that can be used in any setting where the loss function $f(d_n; \theta)$ is twice continuously differentiable in $\theta$. They define a quantity-of-interest, $\phi(\theta, w)$, to be a scalar related to the conclusion of a data analysis, which one is concerned about observing a change in upon dropping a very small fraction of data. Common quantities of interest in a data analysis include the sign or significance of a regression coefficient.

Specifically, they linearize the quantity-of-interest as a function of the data-weights

$$\phi^{\mathrm{lin}}(w) = \phi(1_N) + \sum_{n=1}^{N}(w_n - 1)\frac{\partial\phi(w)}{\partial w_n}\Big|_{w=1_N}. \tag{8}$$

The derivative $\frac{\partial\phi(w)}{\partial w_n}\big|_{w=1_N}$ is known as the *influence score* of data point $n$ for $\phi$ at $1_N$.

Although the AMIP methodology has been developed and used for general quantities of interest, $\phi(\hat{\theta}(w), w)$, we focus on the change in sign of a specified regression coefficient; thus, $\phi(\hat{\theta}(w), w) = -\hat{\theta}_p(w)$. Under the general setting where $\hat{\theta}(1_N)$ is the solution to the equation $(\sum_{n=1}^{N}\nabla_\theta f(\hat{\theta}(1_N), d_n)) = 0_P$ (which is the case in our setup as $\hat{\theta}(1_N)$ is a minimizer of a loss function) the implicit function theorem allows us to transform a derivative in $w$ space into a derivative in $\theta$ space

$$\frac{\partial\hat{\theta}(\vec{w})}{\partial w_n}\Big|_{\vec{w}=1_N} = -H(1_N)^{-1}\nabla_\theta f(\hat{\theta}(1_N), d_n) \tag{9}$$

where $H(w) := \sum_{n=1}^{N} w_n \nabla_\theta^2 f(\hat{\theta}(1_N), d_n)$ is the Hessian of the weighted loss. See Broderick et al. (2020) for a detailed derivation of Equation (9). In the context of OLS, we consider the squared error loss, $f(\hat{\theta}(1_N), d_n) = (y_n - \hat{\theta}(1_N)^\top x_n)^2$. The gradient for this loss is $\nabla_\theta f(\hat{\theta}(1_N), d_n) = 2x_n(y_n - \hat{\theta}(1_N)^\top x_n)$, and the Hessian is $\nabla_\theta^2 f(\hat{\theta}(1_N), d_n) = 2x_n x_n^\top$ (Belsley et al., 1980). Thus, from Equation (9), we get the expression

for the influence score for data point $n$,

$$\frac{\partial \hat{\theta}(\vec{w})}{\partial w_n}\Big|_{\vec{w}=1_N} = - \underbrace{(\mathbf{X}^\top \mathbf{X})^{-1} x_n}_{\text{leverage-like term}} \underbrace{(y_n - \hat{\theta}(1_N)^\top x_n)}_{\text{residual term}}. \tag{10}$$

Let $e_p$ be the $p$th standard basis vector. Then the linear approximation in the setting where the quantity of interest is the sign of the $p$th regression coefficient becomes

$$\hat{\theta}_p^{\text{lin}}(w) = \hat{\theta}_p(1_N) + e_p^\top H(1_N)^{-1} \sum_{n=1}^{N} (w_n - 1) \nabla_\theta f(\hat{\theta}(1_N), d_n). \tag{11}$$

## B.2 Additive One-step Newton approximation

Past work has proposed using the One-step Newton (1sN) approximation to estimate how much dropping a pre-defined subset of data changes the loss, for general losses (Beirami et al., 2017; Sekhari et al., 2021; Koh et al., 2019; Ghosh et al., 2020). When we simultaneously consider (a) OLS linear regression and (b) our particular (effect-size) quantity of interest, Additive One-step Newton is equivalent to the Additive One-Exact approximation. So for the experiments in this work, there is no distinction. While Park et al. (2023) first proposed the special case of Additive One-step Newton for logistic regression (in the context of data attribution), we develop a general form of the approximation below. We hope the more general form will prove useful in future works on worst-case data dropping beyond OLS; in particular, in models that are expensive to run, a practitioner might be unwilling to incur the cost of running Additive One-Exact.

The One-step Newton approximation works by optimizing a second-order Taylor expansion to the loss around $w = 1_N$. In the case where we must search for the worst-case data subset to drop, we approximate $\hat{\theta}(w)$ with

$$\hat{\theta}^{\text{1sN}}(w) := \hat{\theta}(1_N) + H(w)^{-1} \sum_{n=1}^{N} (w_n - 1) \nabla_\theta f(\theta(1_N), d_n). \tag{12}$$

The One-step Newton approximation allows us to approximate $\hat{\theta}(w) = \arg\min \sum_{n=1}^{N} w_n f(\theta, d_n)$ with a second-order Taylor series expansion (in $\theta$) centered at the estimate for the full data, $\hat{\theta}(1_N)$.

$$
\begin{aligned}
\sum_{n=1}^{N} w_n f(\theta, d_n) \approx{}& f(\hat{\theta}(1_N), d_n) + \sum_{n=1}^{N} w_n \nabla f(\hat{\theta}(1_N), d_n)(\theta - \hat{\theta}(1_N)) \\
& + \frac{1}{2}(\theta - \hat{\theta}(1_N))^\top \sum_{n=1}^{N} w_n \nabla^2 f(\hat{\theta}(1_N), d_n)(\theta - \hat{\theta}(1_N))
\end{aligned}
\tag{13}
$$

In order to solve for $\arg\min \sum_{n=1}^{N} w_n f(\theta, d_n)$, we can minimize the quadratic approximation to get

$$\hat{\theta}^{\text{1sN}}(w) = \hat{\theta}(1_N) + \Big( \sum_{n=1}^{N} w_n \nabla^2 f(\hat{\theta}(1_N), d_n) \Big)^{-1} \sum_{n=1}^{N} w_n \nabla f(\hat{\theta}(1_N), d_n). \tag{14}$$

In the recent machine learning literature, this One-step Newton approximation (Equation (14)) has been proposed to estimate the effect of dropping known subsets of data (Beirami et al., 2017; Sekhari et al., 2021; Koh et al., 2019; Ghosh et al., 2020) in the context of general twice-differentiable losses.

In the setting of simple linear regression, the One-step Newton approximation gives the exact solution to the reweighted OLS estimate of a regression coefficient (Pregibon, 1981, Equation 3). Let $\mathbf{X} \in \mathbb{R}^{N \times P}$ denote the design matrix and $y \in \mathbb{R}^N$ denote the response vector. Let $S$ denote the dropped set (i.e., the observations indexed by $S$ in the design matrix and response vector) and $\backslash S$ denote its complement. Let $\hat{\theta}^{\text{1sN}}(w)$ denote the One-step Newton approximation of $\hat{\theta}(w)$ given in Equation (12).

$$
\begin{aligned}
\hat{\theta}^{\text{1sN}}(w) &= (\mathbf{X}^\top \mathbf{X})^{-1} \mathbf{X}^\top y + (\mathbf{X}_{\backslash S}^\top \mathbf{X}_{\backslash S})^{-1}(\mathbf{X}_S^\top y_S - \mathbf{X}_S^\top \mathbf{X}_S(\mathbf{X}^\top \mathbf{X})^{-1}\mathbf{X}^\top y) \\
&= (\mathbf{X}^\top \mathbf{X})^{-1} \mathbf{X}^\top y - (\mathbf{X}_{\backslash S}^\top \mathbf{X}_{\backslash S})^{-1}(\mathbf{X}_{\backslash S}^\top y_{\backslash S} - \mathbf{X}_{\backslash S}^\top \mathbf{X}_{\backslash S}(\mathbf{X}^\top \mathbf{X})^{-1}\mathbf{X}^\top y) \\
&= (\mathbf{X}_{\backslash S}^\top \mathbf{X}_{\backslash S})^{-1} \mathbf{X}_{\backslash S}^\top y_{\backslash S}
\end{aligned}
\tag{15}
$$

such that

$$\hat{\theta}(w) - \hat{\theta}^{\text{1sN}}(w) = (\mathbf{X}_{\backslash S}^{\top}\mathbf{X}_{\backslash S})^{-1}\mathbf{X}_{\backslash S}^{\top}y_{\backslash S} - (\mathbf{X}_{\backslash S}^{\top}\mathbf{X}_{\backslash S})^{-1}\mathbf{X}_{\backslash S}^{\top}y_{\backslash S} = 0. \tag{16}$$

The One-step Newton approximation has not been proposed in the context of the Maximum Influence Perturbation problem because (unlike influence scores) the approximation is non-additive. This non-additivity precludes the fast solution of approximating the Most Influential Set by ranking and taking a sum of the top individual scores (Broderick et al., 2020). As a solution, we adapt ideas from the AMIP to consider an approximation that uses a sum of One-step Newton scores for leaving out individual data points (Equation (17)). We refer to this approximation, which was first proposed for the special case of logistic regression by Park et al. (2023) (in the context of data attribution), as *Additive One-step Newton (Add-1sN)*.

$$\hat{\theta}^{\text{Add-1sN}}(w) = \hat{\theta}(1_N) + \sum_{n \in S}\left(\left(\sum_{\substack{n'=1, \\ n' \neq n}}^{N}\nabla^2 f(\hat{\theta}(1_N), d_{n'})\right)^{-1}\nabla f(\hat{\theta}(1_N), d_n)\right). \tag{17}$$

Add-1sN applies to general differentiable losses, though we continue to focus on a quantity of interest equal to a particular parameter value (so our approximation does not include a decision based on statistical significance).

A promising direction for future research is to extend the Add-1sN to more general quantities of interest and $Z$-estimators. Ideally, such extensions would be automatic through autodiff, similar to the approach taken for the computation of the AMIP (see Broderick et al. (2020)). We anticipate that techniques developed for the AMIP can aid in extending the Add-1sN to more general quantities of interest.

### B.3   Analytic expressions for the error of additive approximations

As the approximation methods presented in Section 3 are local approximations based on removing individual observations, errors may accrue when there exists subsets of points with high joint influence measures but low individual influence measures.

In linear regression, we can formalize this intuition by looking at an analytic expression for the OLS estimator of the $p$th regression coefficient, $\theta$. Let $e_p \in \mathbb{R}^d$ denote the $p$th standard basis vector. Let $\mathbf{X} \in \mathbb{R}^{N \times P}$ denote the design matrix and $y \in \mathbb{R}^N$ denote the response vector. Let $S$ denote the dropped set (i.e., the observations indexed by $S$ in the design matrix and response vector) and $\backslash S$ denote its complement. Let $\hat{\theta}^{\text{IF}}(w)$ denote the influence function approximation of $\hat{\theta}$ given in Equation (11) and let $\hat{\theta}^{\text{Add-1Exact}}(w)$ denote the approximation given in Equation (12).

Let $\mathbf{X}_{-n}$ denote the design matrix leaving out data point $n$, $x_n \in \mathbb{R}^d$ denote the $x$ value of the $n$th data point, and $r_n = (y_n - \hat{\theta}(1_N)^{\top}x_n)$ denote the residual value of the $n$th data point.

The error incurred by AMIP can be written as

$$\begin{aligned}
\hat{\theta}^{\text{AMIP}}(w) - \hat{\theta}(w) &= \sum_{n \in S}e_p^{\top}(\mathbf{X}^{\top}\mathbf{X})^{-1}x_n r_n - \sum_{n \in S}e_p^{\top}(\mathbf{X}_{\backslash S}^{\top}\mathbf{X}_{\backslash S})^{-1}x_n r_n \\
&= e_p^{\top}((\mathbf{X}^{\top}\mathbf{X})^{-1} - (\mathbf{X}_{\backslash S}^{\top}\mathbf{X}_{\backslash S})^{-1})\sum_{n \in S}x_n r_n \\
&= e_p^{\top}((\mathbf{X}^{\top}\mathbf{X})^{-1} - (\mathbf{X}_{\backslash S}^{\top}\mathbf{X}_{\backslash S})^{-1})\mathbf{X}_S^{\top}r_S
\end{aligned} \tag{18}$$

and the error by Additive One-Exact can be written as

$$\begin{aligned}
\hat{\theta}^{\text{Add-1Exact}}(w) - \hat{\theta}(w) &= \sum_{n \in S}e_p^{\top}(\mathbf{X}_{-n}^{\top}\mathbf{X}_{-n})^{-1}x_n r_n - \sum_{n \in S}e_p^{\top}(\mathbf{X}_{\backslash S}^{\top}\mathbf{X}_{\backslash S})^{-1}x_n r_n \\
&= \sum_{n \in S}e_p^{\top}((\mathbf{X}_{-n}^{\top}\mathbf{X}_{-n})^{-1} - (\mathbf{X}_{\backslash S}^{\top}\mathbf{X}_{\backslash S})^{-1})x_n r_n.
\end{aligned} \tag{19}$$

### B.4 Time complexity supplementals

**AMIP:** The AMIP algorithm can be broken down into four steps: (a) run the data analysis on the full dataset, (b) compute the influence scores for each data point, (c) rank the influence scores, and (d) sum the top $\lfloor \alpha N \rfloor$ scores. In the context of OLS, the cost of step (a) is $O(NP^2 + P^3)$. More generally, we can denote the cost of (a) as being $O(Analysis)$. The cost of (b), computing the influence score for $N$ data points, is $O(NP^2 + P^3)$. Recall that the influence score can be expressed as a Hessian-vector product (see Equation (9)). The Hessian is a matrix of dimension $P \times P$. For $N$ data points, the cost of computing the Hessian is $O(NP^2)$. To invert the Hessian (which is done once in the computation) costs $O(P^3)$. The gradient is a vector of dimension $P$. To compute the gradient for $N$ data points costs $O(NP)$. To multiply the Hessian by the gradient, for $N$ data points, costs $O(NP^2)$. Step (c), finding the top $\lfloor \alpha N \rfloor$ influence scores, costs $O(N \log \alpha N)$.[9] Step (d), the summing of top scores, costs $O(N)$.

In general, the overall cost of running AMIP is $O(Analysis + N \log(\alpha N) + NP^2 + P^3)$. For OLS with an effect size quantity of interest, the cost of running OLS, $O(Analysis)$, is $O(NP^2 + P^3)$. Hence, the cost becomes $O(N \log(\alpha N) + NP^2 + P^3)$.

**Additive One-Exact:** The Additive One-Exact algorithm can be broken down into four steps: (a) run the data analysis on the full dataset, (b) compute the One-exact scores (the exact impact of dropping an individual data point) for each point, (c) rank the One-exact scores, and (d) sum the top $\lfloor \alpha N \rfloor$ scores. The only difference between running this algorithm and running AMIP is in step (b). In the general data analysis setting, the computation of One-Exact scores involves the re-running of data analyses upon dropping each individual point in a dataset, a cost that is $O(N \times Analysis)$. In the setting of OLS, we can take advantage of the One-step Newton update in place of re-running the analysis $N$ times (see Appendix B.2 for more details). Using this rank-one update, the cost of computing One-Exact scores for $N$ data points becomes $O(NP^3 + P^3)$, or simply $O(NP^3)$. Notice that the cost differs from computing influence scores by an extra factor of $P$ (recall that the cost of computing influence scores for AMIP is $O(NP^2 + P^3)$). The improved precision of One-Exact scores over influence scores comes at the cost of this additional factor of $P$; specifically, for Additive One-Exact, the Hessian matrix is reweighted to account for each dropped data point (see Equation (17) in Appendix B.2 for the equation for this approximation) while, for AMIP, the reweighting is omitted.

In general, the overall cost of running Additive One-Exact is $O(N \times Analysis + N \log(\alpha N))$, and the cost specific to OLS with an effect size quantity of interest is $O(N \log(\alpha N) + NP^3)$.

**Greedy AMIP:** The Greedy AMIP algorithm can be broken down into three steps, iterated over $\lfloor \alpha N \rfloor$ times: a) approximate the change (to the quantity of interest) upon dropping each data point individually using an influence function approximation, (b) select the point that results in the biggest approximated change when dropped, and (c) re-run the data analysis. Recall that computing the influence scores for $N$ data points costs $O(NP^2 + P^3)$; iterated for $\lfloor \alpha N \rfloor$ times, step (a) costs $O(\alpha N^2 P^2 + \alpha NP^3)$. To find the top influence score $\lfloor \alpha N \rfloor$ times, step (b) costs $O(\alpha N^2)$. Finally, to re-run the data analysis $\lfloor \alpha N \rfloor$ times with the point dropped, step (c) costs $O(\alpha N \times Analysis)$. In general, running Greedy AMIP costs $O(\alpha N \times Analysis + \alpha N^2 P^2 + \alpha NP^3)$. For OLS with an effect size quantity of interest, the cost of running OLS, $O(Analysis)$, is $O(NP^2 + P^3)$. Thus, the cost of running Greedy AMIP for OLS with an effect size quantity of interest is $O(\alpha N^2 P^2 + \alpha NP^3)$.

**Greedy One-Exact:** The Greedy One-Exact algorithm can be broken down into three steps, iterated over $\lfloor \alpha N \rfloor$ times: a) compute the exact change (to the quantity of interest) upon dropping each data point individually, (b) select the point that results in the biggest change when dropped, and (c) re-run the data analysis. In the general data analysis setting, when iterated for $\lfloor \alpha N \rfloor$ times, computing the exact changes for leaving out each point individually, step (a) costs $O(\alpha N^2 \times Analysis)$. To find the top score costs $O(N)$. Repeated over $\lfloor \alpha N \rfloor$ times, step (b) costs $O(\alpha N^2)$. To re-run the data analysis after dropping the top point, step (c) costs $O(\alpha N \times Analysis)$.

Thus, the general overall cost of running Greedy One-Exact is $O(\alpha N^2 \times Analysis)$. For OLS with an effect size quantity of interest, this cost reduces to $O(\alpha N^2 P^3)$.

---

[9]In the limit, $\lfloor \alpha N \rfloor$ is equivalent to $\alpha N$.

### B.5   Asymptotic Equivalence of $\lfloor \alpha N \rfloor$ and $\alpha N$

The floor function introduces discrete rounding effects that are negligible in the asymptotic regime. We formalize this intuition below.

We begin with the inequality:

$$\alpha N - 1 \leq \lfloor \alpha N \rfloor \leq \alpha N. \tag{20}$$

This inequality holds for all $\alpha \in (0, 1)$ and all $N \in \mathbb{N}$. To simplify the comparison in big-$O$ notation, we seek multiplicative bounds.

Observe that for $N > \frac{2}{\alpha}$, we have that

$$\alpha N - 1 > \frac{1}{2} \alpha N, \tag{21}$$

which implies,

$$\frac{1}{2} \alpha N \leq \lfloor \alpha N \rfloor \leq \alpha N. \tag{22}$$

Hence, for sufficiently large $N$, the floor term $\lfloor \alpha N \rfloor$ is sandwiched between two constants times $\alpha N$, so we conclude:

$$\lfloor \alpha N \rfloor = O(\alpha N) \quad \text{and} \quad \alpha N = O(\lfloor \alpha N \rfloor). \tag{23}$$

Thus, $\lfloor \alpha N \rfloor$ and $\alpha N$ are asymptotically equivalent up to constant factors.

## C   Failure modes supplementals

### C.1   Tables demonstrating results of approximation methods

Table 1 shows that, in the One Outlier example, AMIP fails both with and without re-run. While there exists one point (in black) that can change the sign of the regression coefficient, the method reports that no subsets of size one exist that can change the sign (a failure without re-run). Upon removal of the point suggested by AMIP (which is a red point) and refitting the model, we still do not see a change in sign (a failure with re-run). Greedy AMIP also faces a failure with re-run because the sign does not change upon refitting after we remove the point identified by the algorithm. In this example, Additive One-Exact and Greedy One-Exact succeed. Of the OLS-specific algorithms, NetApprox and FH-Gurobi (warm-start) also succeed, while integral FH-Gurobi (without warm-start) fails to identify a subset of size one that can change the sign.

Table 2 shows that in the Simpson's Paradox example, AMIP and Additive One-Exact fail both with and without re-run. While there exists a group of ten points ($\alpha = 0.01$) (specifically, the group of points in black) such that, upon removal, the sign of the regression coefficient changes from positive to negative, both AMIP and Additive One-Exact report that no such subset of this size or smaller exists. The sign also does not change upon refitting, after removing the points identified by the algorithms. In this example, the greedy versions of both approximations succeed. Of the mathematical programming algorithms, NetApprox and FH-Gurobi (warm-start) also succeed, while integral FH-Gurobi (without warm-start) fails to identify a subset of size 10 that can change the sign.

Similar to the Simpson's Paradox example, Table 3 shows that in the Poor Conditioning example, AMIP and Additive One-Exact fail both with and without re-run, while the greedy versions of both approximations succeed. Again, of the mathematical programming algorithms, NetApprox and FH-Gurobi (warm-start) succeed while integral FH-Gurobi (without warm-start) fails to identify a subset of size 10 that can change the sign.

Table 4 through Table 8 display the performance of the data-dropping approximations on real-world data sets.

Table 1: Performance of methods under the One Outlier example, where an outlier is placed at (X, Y) = (1e6, 1e6). We know that there exists a subset (one data point!) such that, upon removal, the sign of the regression coefficient changes from positive (1.000) to negative (-1.000). Hence, $\alpha = \frac{1}{N}$ is sufficient to lead to a failure mode. The "Predicted Estimate" column shows the estimate predicted by the approximation algorithm, and the "Refit Estimate" column shows the result of refitting the model after removing the approximate Most Influential Subset specified by the algorithm. The "Points Dropped" column shows the number of red (R) and black (B) points that the algorithm drops. The values highlighted in green indicate that the algorithm succeeded. Non-highlighted values under "Predicted Estimate" indicate a failure without re-run, while non-highlighted values under "Refit Estimate" indicate a failure with re-run.

| Method | Predicted Estimate | Refit Estimate | Points Dropped |
|---|---|---|---|
| Removing Population A | — | -1.000 | (R: 0, B: 1) |
| AMIP | 0.999 | 0.999 | (R: 1, B: 0) |
| Additive One-Exact | -1.000 | -1.000 | (R: 0, B: 1) |
| Greedy AMIP | — | 0.999 | (R: 1, B: 0) |
| Greedy One-Exact | — | -1.000 | (R: 0, B: 1) |
| NetApprox | — | -1.000 | (R: 0, B: 1) |
| FH-Gurobi | — | — | (R: 62, B: 1) |
| FH-Gurobi (warm-start) | — | -1.000 | (R: 0, B: 1) |

Table 2: Performance of methods under the Simpson's Paradox example. We know that there exists a subset (namely, the 10 points in Population A ($\alpha = 0.01$) in Figure 1) such that, upon removal, the sign of the regression coefficient changes from positive (0.586) to negative (-0.990). The "Predicted Estimate" column shows the estimate predicted by the approximation algorithm, and the "Refit Estimate" column shows the result of refitting the model after removing the approximate Most Influential Subset specified by the algorithm. The "Points Dropped" column shows the number of red (R) and black (B) points that the algorithm drops. The values highlighted in green indicate that the algorithm succeeded. Non-highlighted values under "Predicted Estimate" indicate a failure without re-run, while non-highlighted values under "Refit Estimate" indicate a failure with re-run.

| Method | Predicted Estimate | Refit Estimate | Points Dropped |
|---|---|---|---|
| Removing Population A | — | -0.990 | (R: 0, B: 10) |
| AMIP | 0.462 | 0.279 | (R: 2, B: 8) |
| Additive One-Exact | 0.456 | 0.279 | (R: 2, B: 8) |
| Greedy AMIP | — | -0.990 | (R: 0, B: 10) |
| Greedy One-Exact | — | -0.990 | (R: 0, B: 10) |
| NetApprox | — | -0.990 | (R: 0, B: 10) |
| FH-Gurobi | — | — | (R: 112, B: 10) |
| FH-Gurobi (warm-start) | — | -0.990 | (R: 0, B: 10) |

Table 3: Performance of methods under the Poor Conditioning example. We know that there exists a subset (namely, the 10 points in Population A ($\alpha = 0.01$) in Figure 1) such that, upon removal, the sign of the regression coefficient changes from positive (8.452) to negative (-1.049). The "Predicted Estimate" column shows the estimate predicted by the approximation algorithm, and the "Refit Estimate" column shows the result of refitting the model after removing the approximate Most Influential Subset specified by the algorithm. The "Points Dropped" column shows the number of red (R) and black (B) points that the algorithm drops. The values highlighted in green indicate that the algorithm succeeded. Non-highlighted values under "Predicted Estimate" indicate a failure of type (i), while non-highlighted values under "Refit Estimate" indicate a failure of type (ii).

| Method | Predicted Estimate | Refit Estimate | Indices Dropped |
|---|---|---|---|
| Removing Population A | — | -1.049 | (R: 0, B: 10) |
| AMIP | 6.724 | 5.376 | (R:5, B:5) |
| Additive One-Exact | 6.667 | 5.376 | (R: 3, B: 7) |
| Greedy AMIP | — | -1.049 | (R: 0, B: 10) |
| Greedy One-Exact | — | -1.049 | (R: 0, B: 10) |
| NetApprox | — | -1.049 | (R: 0, B: 10) |
| FH-Gurobi | — | — | (R: 991, B: 10) |
| FH-Gurobi (warm-start) | — | -1.049 | (R: 0, B: 10) |

Table 4: Performance of methods on the mouse brain single-cell analysis data set, where $\alpha = 1\%$ is sufficient to lead to a failure mode. The "Predicted Estimate" column shows the estimate predicted by the approximation algorithm, and the "Refit Estimate" column shows the result of refitting the model after removing the approximate Most Influential Subset specified by the algorithm. The "Number Dropped" column shows the number of points that the algorithm drops. The values highlighted in green indicate that the algorithm succeeded. Non-highlighted values under "Predicted Estimate" indicate a failure of type (i), while non-highlighted values under "Refit Estimate" indicate a failure of type (ii).

| Method | Predicted Estimate | Refit Estimate | Number Dropped |
|---|---|---|---|
| AMIP | 0.451 | 0.355 | 656 |
| Additive 1Exact | 0.4511 | 0.355 | 656 |
| Greedy AMIP | — | -0.003 | 172 |
| Greedy 1Exact | — | -0.003 | 172 |
| NetApprox | — | -0.002 | 656 |
| FH-Gurobi | — | 0.000 | 1712 |
| FH-Gurobi (warm-start) | — | 0.002 | 172 |

Table 5: Performance of methods on the Ames Housing data set, where $\alpha = 1\%$ is sufficient to lead to a failure mode. The "Predicted Estimate" column shows the estimate predicted by the approximation algorithm, and the "Refit Estimate" column shows the result of refitting the model after removing the approximate Most Influential Subset specified by the algorithm. The "Number Dropped" column shows the number of points that the algorithm drops. The values highlighted in green indicate that the algorithm succeeded. Non-highlighted values under "Predicted Estimate" indicate a failure of type (i), while non-highlighted values under "Refit Estimate" indicate a failure of type (ii).

| Method | Predicted Estimate | Refit Estimate | Number Dropped |
|---|---|---|---|
| AMIP | 72.010 | 55.845 | 14 |
| Additive 1Exact | 55.844 | 55.845 | 14 |
| Greedy AMIP | — | -15.09 | 4 |
| Greedy 1Exact | — | -15.09 | 4 |
| NetApprox | — | -14.11 | 14 |
| FH-Gurobi | — | -15.05 | 5 |
| FH-Gurobi (ws) | — | -15.05 | 5 |

Table 6: Performance of methods on the bird morphometrics data set, where $\alpha = 1\%$ is sufficient to lead to a failure mode. The "Predicted Estimate" column shows the estimate predicted by the approximation algorithm, and the "Refit Estimate" column shows the result of refitting the model after removing the approximate Most Influential Subset specified by the algorithm. The "Number Dropped" column shows the number of points that the algorithm drops. The values highlighted in green indicate that the algorithm succeeded. Non-highlighted values under "Predicted Estimate" indicate a failure of type (i), while non-highlighted values under "Refit Estimate" indicate a failure of type (ii).

| Method | Predicted Estimate | Refit Estimate | Number Dropped |
|---|---|---|---|
| AMIP | -0.401 | 0.399 | 1 |
| Additive 1Exact | 0.399 | 0.399 | 1 |
| Greedy AMIP | — | 0.399 | 1 |
| Greedy 1Exact | — | 0.399 | 1 |
| NetApprox | — | -0.719 | 1 |
| FH-Gurobi | — | 0.410 | 4 |
| FH-Gurobi (warm-start) | — | 0.554 | 4 |

Table 7: Performance of methods on the photosynthesis measurements data set, where all methods succeed with refit at $\alpha = 0.0036$. The "Predicted Estimate" column shows the estimate predicted by the approximation algorithm, and the "Refit Estimate" column shows the result of refitting the model after removing the approximate Most Influential Subset specified by the algorithm. The "Number Dropped" column shows the number of points that the algorithm drops. The values highlighted in green indicate that the algorithm succeeded. Non-highlighted values under "Predicted Estimate" indicate a failure of type (i), while non-highlighted values under "Refit Estimate" indicate a failure of type (ii).

| Method | Predicted Estimate | Refit Estimate | Number Dropped |
|---|---|---|---|
| AMIP | -0.0117 | -0.0038 | 2 |
| Additive 1Exact | -0.0129 | -0.00377 | 2 |
| Greedy AMIP | — | -0.00377 | 2 |
| Greedy 1Exact | — | -0.00377 | 2 |
| NetApprox | — | -0.00377 | 2 |
| FH-Gurobi | — | -0.00377 | 2 |
| FH-Gurobi (warm-start) | — | -0.00377 | 2 |

Table 8: Performance of methods on the forestry data set, where all methods succeed with refit at $\alpha = 1\%$. The "Predicted Estimate" column shows the estimate predicted by the approximation algorithm, and the "Refit Estimate" column shows the result of refitting the model after removing the approximate Most Influential Subset specified by the algorithm. The "Number Dropped" column shows the number of points that the algorithm drops. The values highlighted in green indicate that the algorithm succeeded. Non-highlighted values under "Predicted Estimate" indicate a failure of type (i), while non-highlighted values under "Refit Estimate" indicate a failure of type (ii).

| Method | Predicted Estimate | Refit Estimate | Number Dropped |
|---|---|---|---|
| AMIP | 0.0167 | -0.0115 | 1 |
| Additive 1Exact | -0.0115 | -0.0115 | 1 |
| Greedy AMIP | — | -0.0115 | 1 |
| Greedy 1Exact | — | -0.0115 | 1 |
| NetApprox | — | -0.0115 | 1 |
| FH-Gurobi | — | -0.0115 | 1 |
| FH-Gurobi (warm-start) | — | -0.0115 | 1 |

### C.2 Failure modes without an intercept term

In this section, we find that fitting without the intercept does not significantly affect the numerical results in Section 4 and that most of the same failure modes still hold.

The one exception to this observation is that, with an intercept term, AMIP with re-run (which is the same as Greedy-AMIP for dropping one data point) fails in the one-outlier example with an intercept term, but without an intercept term, it succeeds. Here, we note that for the case when $P = 1$ and no intercept term, the limiting expression for the arbitrary non-outlier point discussed in Proposition 4.3 (Equation (6)) is always 0 because $e_p$ and $v$ are collinear by design. In more than 1 dimension, however, this collinearity need no longer hold, and so Equation (6) may indeed converge to a non-zero constant, which may again result in failure modes with re-run without an intercept term.

**One-Outlier example.** Upon fitting OLS without an intercept, the coefficient fit on the full dataset is 1.000. The fit with the black-dot points removed is $-1.033$. The removal of the intercept has negligible effects on the numerical results of the algorithms, with the exception of AMIP/Greedy-AMIP "Refit Estimate" (which was 0.999 with an intercept and is $-1.033$ without an intercept) (Table 1).

Table 9: Performance of methods under the One Outlier example, where an outlier is placed at (X, Y) = (1e6, 1e6). We know that there exists a subset (one data point!) such that, upon removal, the sign of the regression coefficient changes from positive (1.000) to negative (−1.033). Hence, $\alpha = \frac{1}{N}$ is sufficient to lead to a failure mode. The "Predicted Estimate" column shows the estimate predicted by the approximation algorithm, and the "Refit Estimate" column shows the result of refitting the model after removing the approximate Most Influential Subset specified by the algorithm. The "Points Dropped" column shows the number of red (R) and black (B) points that the algorithm drops. The values highlighted in green indicate that the algorithm succeeded. Non-highlighted values under "Predicted Estimate" indicate a failure without re-run, while non-highlighted values under "Refit Estimate" indicate a failure with re-run.

| Method | Predicted Estimate | Refit Estimate | Points Dropped |
|---|---|---|---|
| Removing Population A | — | -1.033 | (R: 0, B: 1) |
| AMIP | 1.000 | -1.033 | (R: 0, B: 1) |
| Additive One-Exact | -1.033 | -1.033 | (R: 0, B: 1) |
| Greedy AMIP | — | -1.033 | (R: 0, B: 1) |
| Greedy One-Exact | — | -1.000 | (R: 0, B: 1) |
| NetApprox | — | -1.033 | (R: 0, B: 1) |
| FH-Gurobi | — | -1.033 | (R: 0, B: 1) |
| FH-Gurobi (warm-start) | — | -1.033 | (R: 0, B: 1) |

**Simpon's Paradox.** When we do not include an intercept, the coefficient fit on the full dataset is 0.516. The fit with the black-dot points removed is −0.990. The removal of the intercept has negligible effects on the numerical results of the additive and greedy algorithms (Table 2). However, FH-Gurobi (warm-start) now presents an additional failure.

Table 10: Performance of methods under the Simpson's Paradox example. We know that there exists a subset (namely, the 10 points in Population A ($\alpha = 0.01$) in Figure 1) such that, upon removal, the sign of the regression coefficient changes from positive (0.516) to negative (−0.990). The "Predicted Estimate" column shows the estimate predicted by the approximation algorithm, and the "Refit Estimate" column shows the result of refitting the model after removing the approximate Most Influential Subset specified by the algorithm. The "Points Dropped" column shows the number of red (R) and black (B) points that the algorithm drops. The values highlighted in green indicate that the algorithm succeeded. Non-highlighted values under "Predicted Estimate" indicate a failure without re-run, while non-highlighted values under "Refit Estimate" indicate a failure with re-run.

| Method | Predicted Estimate | Refit Estimate | Points Dropped |
|---|---|---|---|
| Removing Population A | — | -0.990 | (R: 0, B: 10) |
| AMIP | 0.462 | 0.278 | (R: 2, B: 8) |
| Additive One-Exact | 0.455 | 0.278 | (R: 2, B: 8) |
| Greedy AMIP | — | -0.990 | (R: 0, B: 10) |
| Greedy One-Exact | — | -0.990 | (R: 0, B: 10) |
| NetApprox | — | -0.990 | (R: 0, B: 10) |
| FH-Gurobi | — | — | (R: 810, B: 10) |
| FH-Gurobi (warm-start) | — | 0.514 | (R: 0, B: 1) |

**Poor Conditioning.** When we do not include an intercept, the coefficient fit on the full dataset is 7.405. The fit with the black-dot points removed is −1.042. The removal of the intercept has negligible effects on all numerical results (Table 3).

Table 11: Performance of methods under the Poor Conditioning example. We know that there exists a subset (namely, the 10 points in Population A ($\alpha = 0.01$) in Figure 1) such that, upon removal, the sign of the regression coefficient changes from positive (8.452) to negative (-1.049). The "Predicted Estimate" column shows the estimate predicted by the approximation algorithm, and the "Refit Estimate" column shows the result of refitting the model after removing the approximate Most Influential Subset specified by the algorithm. The "Points Dropped" column shows the number of red (R) and black (B) points that the algorithm drops. The values highlighted in green indicate that the algorithm succeeded. Non-highlighted values under "Predicted Estimate" indicate a failure of type (i), while non-highlighted values under "Refit Estimate" indicate a failure of type (ii).

| Method | Predicted Estimate | Refit Estimate | Indices Dropped |
|---|---|---|---|
| Removing Population A | — | -1.049 | (R: 0, B: 10) |
| AMIP | 6.616 | 5.376 | (R:3, B:7) |
| Additive One-Exact | 6.548 | 5.376 | (R: 3, B: 7) |
| Greedy AMIP | — | `-1.049` | `(R: 0, B: 10)` |
| Greedy One-Exact | — | `-1.042` | `(R: 0, B: 10)` |
| NetApprox | — | `-1.049` | `(R: 0, B: 10)` |
| FH-Gurobi | — | — | (R: 991, B: 10) |
| FH-Gurobi (warm-start) | — | `-1.049` | `(R: 0, B: 10)` |

## C.3 Failure with and without re-run for FH-Gurobi

In this section, we discuss the distinction between failure with and without re-run for FH-Gurobi. Technically, the FH-Gurobi algorithms do not always require the user to re-run their data analysis with the suggested points dropped. However, the two failure modes are still equivalent for these approximations. To see the equivalence, assume that the data are non-robust to data dropping—specifically, that there exists some set of size $\lfloor \alpha N \rfloor$ that we can drop to change conclusions. If the approximation returns a set of size no greater than $\lfloor \alpha N \rfloor$, then approximation calls for users to re-run the data analysis with the identified subset dropped, in order to determine whether conclusion (e.g., the sign of an effect size) indeed changes. Thus, the two failure modes are equivalent in this setting. If the method returns a set of size greater than $\lfloor \alpha N \rfloor$, then a failure has occurred because no set of size at most $\lfloor \alpha N \rfloor$ was found, so the user does not have to re-run their data analysis. If they had re-run their analysis without this subset, the set is still greater than size $\lfloor \alpha N \rfloor$, which still implies that the approximation failed. Thus, both failure types are again equivalent.

## C.4 Successful examples in real-world data

To get a sense of examples in which data-dropping approximations do succeed in the real world, we provide some real-world data sets that are non-robust to small-fraction data dropping, yet for which all methods succeed.

### C.4.1 Plants

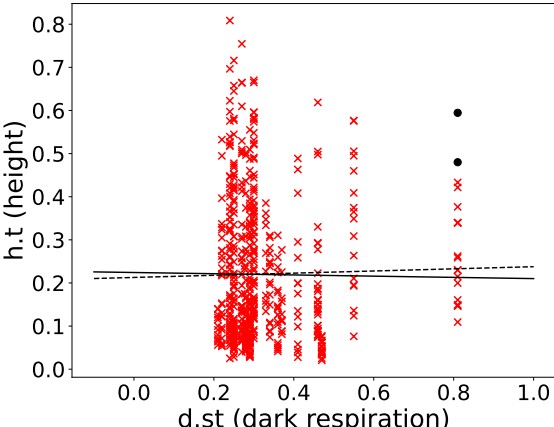

Figure 4: Plant photosynthesis data. Dashed line indicates the fit to the full data while solid line indicates the fit with the 2 black-dot points removed.

**Setup.** This data set is taken from an ecological study on the plastic phenotypic response to light of shrubs from a Panamanian rainforest Valladares et al. (2000). We consider a 1D linear regression (with intercept) of height on dark respiration.

**Experimental results.** We know that there exists a subset of size 2 points (0.36% of the data) that we can drop to change the sign of the regression coefficient from positive (0.0249) to negative ($-0.014$). In this example, all methods succeed at identifying the two points that, when dropped, change the sign. Thus, all methods succeed at $\alpha = 1\%$.

### C.4.2 Forestry

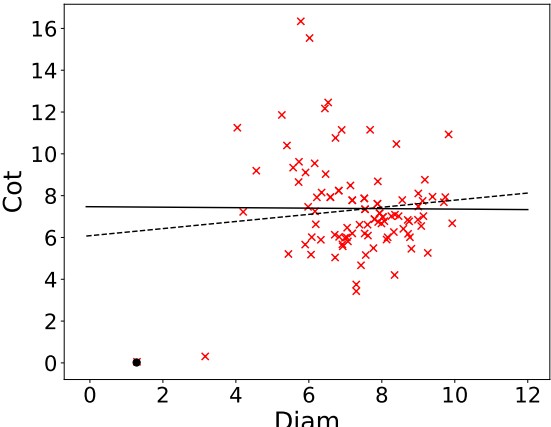

Figure 5: Forestry data. Dashed line indicates the fit to the full data while solid line indicates the fit with the 1 black-dot point removed. Though it is not visually apparent from the plot, we note here that the black-dot point overlays one other red-cross point (with slightly different $x$ and $y$ values).

**Setup.** This data set is taken from a database recording various characteristics of woody plants (age, leaf size, leaf mass per area, wood density, nitrogen content of leaves and wood), as well as information about their growing environment (location, light, experimental treatment, vegetation type) (York, 2016). To ensure that 1 data point is around 1% (this data set has a total of 92 points with non-missing values), we augmented the data by repeating the 9 red-cross points closest in euclidean distance to the mean of the red-cross points. We consider a 1D linear regression (with intercept) of the two variables Cot (plausibly a component of tree biomass or a specific measurement related to tree structure[10]) on Diam (stem diameter).

**Experimental results.** We know that there exists 1 point in 101 (0.99% of the data) that we can drop to change the sign of the regression coefficient from positive (0.170) to negative ($-0.0115$). In this example, all methods succeed at identifying the one black-dot point that, when dropped, change the sign. Thus, all methods succeed at $\alpha = 1\%$.

### C.5 On the relationship between leverage scores and failures of additive approximations

A common theme across the multi-outlier failure modes presented in Section 4.3 is that the leverage scores of the outlier points are extremely large (see Figure 6).

---

[10]variable definition not explicitly stated in the paper

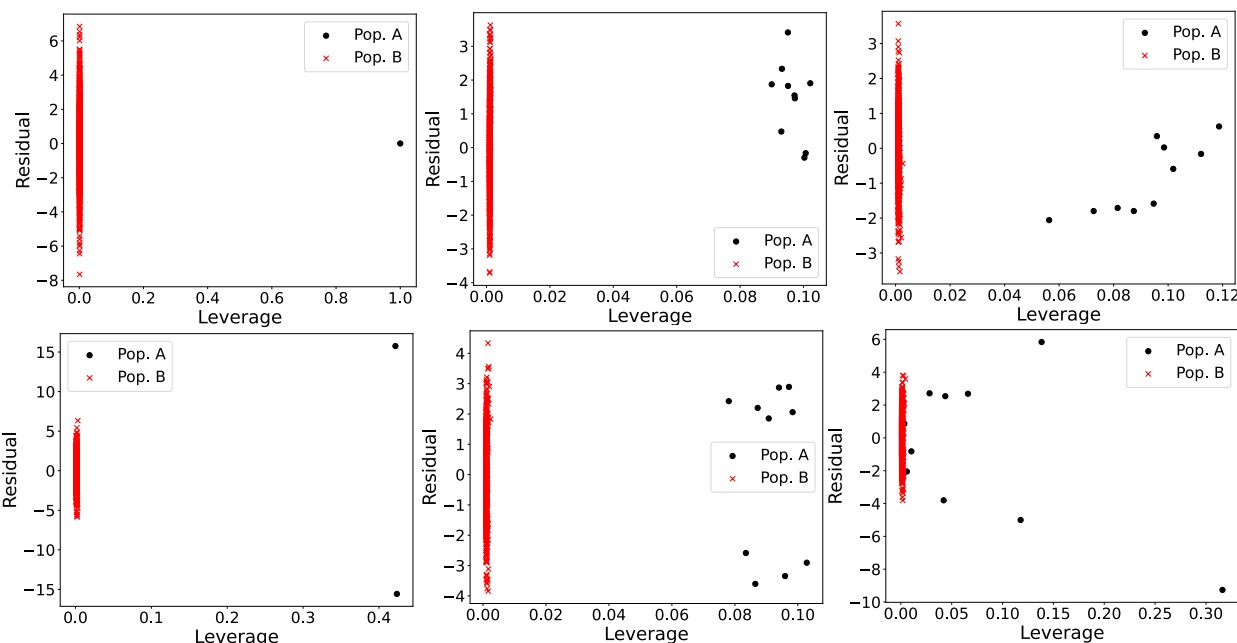

Figure 6: Plots of Residual vs. Leverage: one-outlier (top left), Simpson's paradox (top middle), poor conditioning (top right), two outliers (bottom left), two-outlier groups (bottom middle), two populations (bottom right). In all surfaced failure modes, the leverage values of each black-dot point is larger than the leverage values of each red-cross point.

Data-dropping is, in general, non-additive (Belsley et al., 1980; Gray & Ling, 1984). This fact becomes more apparent when the data points being dropped have high leverage scores. Data points with high leverage scores may interact with other data points in highly non-linear ways, leading to pronounced non-additivity in data-dropping (Gray & Ling, 1984; Lawrance, 1995).

Recall, the leverage score for data point $n$ is the $n$th diagonal entry of the least-squares projection matrix (also known as the Hat matrix). This value can be interpreted as the degree by which the $n$th observation impacts the $n$th fitted value (see Equation (24)) (Belsley et al., 1980). Similarly, the $(n, m)$th off-diagonal entry of the Hat matrix, $h_{nm}$, can be interpreted as the degree by which the $n$th observation impacts the $m$th fitted value (Gray & Ling, 1984). Thus, the entries of the Hat matrix tell us important information about the second-order interaction effects between pairs of data points, information that the additive approximations fail to capture.

$$h_{nn} = \frac{\partial \hat{y}_n}{\partial y_n} \tag{24}$$

The leverage score, $h_{nn}$, bounds the off-diagonal elements of the Hat matrix, $h_{nm}$. These off-diagonal elements capture information about pairwise interactions between points (see Proposition C.1). When $h_{nm}$ is large, additive approximations become poor approximations. This explains why, in all of the surfaced failure modes of Section 4.3, the leverage scores for points in the Most Influential Set are large.

**Proposition C.1.** *Let $x_n \in \mathbb{R}^P$ denote the $n$th column of the design matrix $\mathbf{X} \in \mathbb{R}^{N \times P}$. Let $h_{nm} := x_n^\top (\mathbf{X}^\top \mathbf{X})^{-1} x_m$ denote the entries of the Hat matrix $H := \mathbf{X}(\mathbf{X}^\top \mathbf{X})^{-1} \mathbf{X}^\top$. It follows that*

$$|h_{nm}| \leq \sqrt{h_{nn} h_{mm}}. \tag{25}$$

*Proof.* The Cauchy-Schwarz inequality states that for any vectors $a, b$, in an inner product space,

$$|\langle a, b \rangle| \leq \|a\| \|b\|.$$

Let $a := (\mathbf{X}^\top \mathbf{X})^{-1/2} x_n$ and $b := (\mathbf{X}^\top \mathbf{X})^{-1/2} x_m$. Notice that the entries of the Hat matrix can be written in terms of our defined vectors,

$$h_{nm} = \langle a, b \rangle, h_{nn} = \|a\|^2, h_{mm} = \|b\|^2.$$

Taking square roots,

$$\|a\| = \sqrt{h_{nn}}, \|b\| = \sqrt{h_{mm}}.$$

We thus conclude that

$$|h_{nm}| \leq \sqrt{h_{nn} h_{mm}}.$$

$\square$

### C.6  One-outlier example

#### C.6.1  One-outlier failure mode theory

We saw that both AMIP and Greedy AMIP break down in the one-outlier setting. This is because the influence score of the outlier vanishes as the point approaches infinity in the $x$ and $y$ directions. In Proposition 4.3, we examine the mathematics behind this phenomenon.

**Lemma C.2.** *Let $\lambda \in \mathbb{R}$ and $e_p \in \mathbb{R}^P$ be the pth standard basis vector. Let $x_n \in \mathbb{R}^P$ denote the nth row of the design matrix $\mathbf{X} \in \mathbb{R}^{N \times P}$, and let $y_n \in \mathbb{R}$ denote the nth entry of the response vector $y \in \mathbb{R}^N$. Let $\mathbf{X}_{-1} \in \mathbb{R}^{N-1 \times P}$ denote the design matrix with the 1st row deleted, and let $y_{-1} \in \mathbb{R}^{N-1}$ denote the response vector with the 1st entry deleted. Let $\mathbf{A}_{-1} = \mathbf{X}_{-1}^\top \mathbf{X}_{-1}$ and $b_{-1} = y_{-1}^\top \mathbf{X}_{-1}$. For any $v \in \mathbb{R}^P$ with $\|v\| = 1$ and any constant $c > 0$, let $(x_1, y_1) = (\lambda v, c\lambda)$. For $2 \leq n \leq N$, let $(x_n, y_n) \in \mathbb{R}^P \times \mathbb{R}$ be arbitrary points with the constraint that $\mathbf{X}_{-1}$ has rank $P$. The influence score of data point $(x_1, y_1)$ is,*

$$\left.\frac{\partial \hat{\theta}_p(w)}{\partial w_1}\right|_{w=1_N} = \frac{1}{\lambda^2} \left( \frac{e_p^\top \mathbf{A}_{-1}^{-1} v (c - b_{-1} \mathbf{A}_{-1}^{-1} v)}{\lambda^{-4} + 2\lambda^{-2} v^\top \mathbf{A}_{-1}^{-1} v + (v^\top \mathbf{A}_{-1}^{-1} v)^2} \right). \tag{26}$$

*Proof.* Recall that, for OLS linear regression and the effect size quantity of interest, $\theta_p$, the formula for the influence score of the $n$th data point is

$$\left.\frac{\partial \hat{\theta}_p(w)}{\partial w_n}\right|_{w=1_N} = \underbrace{e_p^\top (\mathbf{X}^\top \mathbf{X})^{-1} x_n}_{\text{leverage-like term}} \underbrace{(y_n - \hat{\theta}^\top x_n)}_{\text{residual term}}. \tag{27}$$

We start by examining the leverage-like term,

$$e_p^\top \left(\mathbf{X}^\top \mathbf{X}\right)^{-1} x_1. \tag{28}$$

Using the Sherman-Morrison formula, the leverage-like term is equivalent to

$$e_p^\top (\mathbf{A}_{-1} + x_1 x_1^\top)^{-1} x_1 = e_p^\top \left( \mathbf{A}_{-1}^{-1} - \frac{\mathbf{A}_{-1}^{-1} x_1 x_1^\top \mathbf{A}_{-1}^{-1}}{1 + x_1^\top \mathbf{A}_{-1}^{-1} x_1} \right) x_1. \tag{29}$$

Substituting $x_1 = \lambda v$ and $y_1 = c\lambda$, this term becomes

$$\frac{\lambda e_p^\top \mathbf{A}_{-1}^{-1} v}{1 + \lambda^2 v^\top \mathbf{A}_{-1}^{-1} v}, \tag{30}$$

which tends to zero as $\lambda \to \infty$.

We next look at the residual term.

The fitted value for $x_1$ is

$$\hat{\theta}^\top x_1 = y^\top \mathbf{X} (\mathbf{X}^\top \mathbf{X})^{-1} x_1. \tag{31}$$

Using the Sherman-Morrison formula, the fitted value can be written as

$$y^\top \mathbf{X} \boldsymbol{A}_{-1}^{-1} x_1 - \frac{y^\top \mathbf{X} \boldsymbol{A}_{-1}^{-1} x_1 x_1^\top \boldsymbol{A}_{-1}^{-1} x_1}{1 + x_1^\top \boldsymbol{A}_{-1}^{-1} x_1}. \tag{32}$$

Through algebraic simplification, the above expression can be written under one fraction,

$$
\begin{aligned}
y^\top \mathbf{X} \boldsymbol{A}_{-1}^{-1} x_1 - \frac{y^\top \mathbf{X} \boldsymbol{A}_{-1}^{-1} x_1 x_1^\top \boldsymbol{A}_{-1}^{-1} x_1}{1 + x_1^\top \boldsymbol{A}_{-1}^{-1} x_1} &= y^\top \mathbf{X} \boldsymbol{A}_{-1}^{-1} x_1 \left( 1 - \frac{x_1^\top \boldsymbol{A}_{-1}^{-1} x_1}{1 + x_1^\top \boldsymbol{A}_{-1}^{-1} x_1} \right) \\
&= \frac{y^\top \mathbf{X} \boldsymbol{A}_{-1}^{-1} x_1}{1 + x_1^\top \boldsymbol{A}_{-1}^{-1} x_1} \\
&= \frac{(y_1 x_1^\top + b_{-1}) \boldsymbol{A}_{-1}^{-1} x_1}{1 + x_1^\top \boldsymbol{A}_{-1}^{-1} x_1}.
\end{aligned}
\tag{33}
$$

Substituting $x_1 = \lambda v \in \mathbb{R}^P$, $y_1 = c\lambda$, we get

$$\frac{(c\lambda^2 v^\top + b_{-1}) \boldsymbol{A}_{-1}^{-1} \lambda v}{1 + \lambda^2 v^\top \boldsymbol{A}_{-1}^{-1} v} = \frac{c\lambda^3 v^\top \boldsymbol{A}_{-1}^{-1} v + \lambda b_{-1} \boldsymbol{A}_{-1}^{-1} v}{1 + \lambda^2 v^\top \boldsymbol{A}_{-1}^{-1} v}. \tag{34}$$

Finally, subtracting the fitted value from $y_1$, the residual is

$$y_1 - \hat{\theta}^\top x_1 = \frac{c\lambda - \lambda b_{-1} \boldsymbol{A}_{-1}^{-1} v}{1 + \lambda^2 v^\top \boldsymbol{A}_{-1}^{-1} v}. \tag{35}$$

Taken together, the influence score of $(x_1, y_1)$ is

$$
\begin{aligned}
e_p^\top (\mathbf{X}^\top \mathbf{X})^{-1} x_1 (y_1 - \hat{\theta}^\top x_1) &= \frac{\lambda^2 (c e_p^\top \boldsymbol{A}_{-1}^{-1} v - e_p^\top \boldsymbol{A}_{-1}^{-1} v b_{-1} \boldsymbol{A}_{-1}^{-1} v)}{1 + 2\lambda^2 v^\top \boldsymbol{A}_{-1}^{-1} v + \lambda^4 (v^\top \boldsymbol{A}_{-1}^{-1} v)^2} \\
&= \frac{1}{\lambda^2} \left( \frac{e_p^\top \boldsymbol{A}_{-1}^{-1} v (c - b_{-1} \boldsymbol{A}_{-1}^{-1} v)}{\lambda^{-4} + 2\lambda^{-2} v^\top \boldsymbol{A}_{-1}^{-1} v + (v^\top \boldsymbol{A}_{-1}^{-1} v)^2} \right).
\end{aligned}
\tag{36}
$$

$\square$

We saw that AMIP failed both with and without re-run in the one-outlier setting.

In the one-outlier case in Section 4.2, we saw that, for sufficiently large $\lambda$, the influence score for the outlier becomes smaller than that of a non-outlier. In Proposition 4.3, we explain this phenomenon more formally.

**Proposition 4.3.** *Choose any $v \in \mathbb{R}^P$ with $\|v\| = 1$ and any constant $c > 0$. Let $(x_1, y_1) = (\lambda v, \lambda c)$. Let $(x_n, y_n)_{n=2}^N$ be any points in $\mathbb{R}^P \times \mathbb{R}$ such that $\mathbf{X}_{-1}$ has rank $P$. Let $\hat{\theta}_p$ denote the $p$th entry of the OLS estimator, $\hat{\theta}$, fit without an intercept. Then, for all $1 \le p \le P$,*

$$\lim_{\lambda \to \infty} \frac{\partial \hat{\theta}_p(w)}{\partial w_1} \bigg|_{w=1_N} = 0, \quad (5) \qquad and \qquad \lim_{\lambda \to \infty} \frac{\partial \hat{\theta}_p(w)}{\partial w_2} \bigg|_{w=1_N} = \frac{st}{(v^\top \boldsymbol{A}_{-1}^{-1} v)^2}, \quad (6)$$

*where $s := (v^\top \boldsymbol{A}_{-1}^{-1} v e_p^\top \boldsymbol{A}_{-1}^{-1} x_2 - e_p^\top \boldsymbol{A}_{-1}^{-1} v v^\top \boldsymbol{A}_{-1}^{-1} x_2)$ and $t := (y_2 v^\top \boldsymbol{A}_{-1}^{-1} v - c v^\top \boldsymbol{A}_{-1}^{-1} x_2 - b_{-1} \boldsymbol{A}_{-1}^{-1} x_2 v^\top \boldsymbol{A}_{-1}^{-1} v + b_{-1} \boldsymbol{A}_{-1}^{-1} v v^\top \boldsymbol{A}_{-1}^{-1} x_2)$.*

*Proof.* Let $e_p \in \mathbb{R}^P$ denote the $p$th standard basis vector. Let $\boldsymbol{A}_{-1} = \mathbf{X}_{-1}^\top \mathbf{X}_{-1}$ and $b_{-1} = y_{-1}^\top \mathbf{X}_{-1}$, where $y_{-1} \in \mathbb{R}^{N-1}$ denotes the response vector with the $n$th entry deleted.

We begin by examining the influence score of $(x_1, y_1)$.

From Lemma C.2 Equation (36), we saw that the influence score of data point $(x_1, y_1)$ is

$$\left. \frac{\partial \hat{\theta}_p(w)}{\partial w_1} \right|_{w=1_N} = \frac{1}{\lambda^2} \left( \frac{e_p^\top \boldsymbol{A}_{-1}^{-1} v (c - b_{-1} \boldsymbol{A}_{-1}^{-1} v)}{\lambda^{-4} + 2\lambda^{-2} v^\top \boldsymbol{A}_{-1}^{-1} v + (v^\top \boldsymbol{A}_{-1}^{-1} v)^2} \right). \tag{37}$$

Taking a limit as $\lambda \to \infty$, this expression goes to zero at rate $O(\frac{1}{\lambda^2})$,

$$\lim_{\lambda \to \infty} \frac{1}{\lambda^2} \left( \frac{e_p^\top \boldsymbol{A}_{-1}^{-1} v (c - b_{-1} \boldsymbol{A}_{-1}^{-1} v)}{\lambda^{-4} + 2\lambda^{-2} v^\top \boldsymbol{A}_{-1}^{-1} v + (v^\top \boldsymbol{A}_{-1}^{-1} v)^2} \right) = 0. \tag{38}$$

We next examine the influence score of $(x_2, y_2)$. We start by examining the leverage-like term,

$$e_p^\top (\mathbf{X}^\top \mathbf{X})^{-1} x_2. \tag{39}$$

Using the Sherman-Morrison formula, this is equivalent to

$$e_p^\top (\boldsymbol{A}_{-1} + x_1 x_1^\top)^{-1} x_2 = e_p^\top \left( \boldsymbol{A}_{-1}^{-1} - \frac{\boldsymbol{A}_{-1}^{-1} x_1 x_1^\top \boldsymbol{A}_{-1}^{-1}}{1 + x_1^\top \boldsymbol{A}_{-1}^{-1} x_1} \right) x_2. \tag{40}$$

Substituting $x_1 = \lambda v$ and combining fractions, the leverage-like term becomes

$$e_p^\top (\mathbf{X}^\top \mathbf{X})^{-1} x_2 = \frac{e_p^\top \boldsymbol{A}_{-1}^{-1} x_2 + \lambda^2 (v^\top \boldsymbol{A}_{-1}^{-1} v e_p^\top \boldsymbol{A}_{-1}^{-1} x_2 - e_p^\top \boldsymbol{A}_{-1}^{-1} v v^\top \boldsymbol{A}_{-1}^{-1} x_2)}{1 + \lambda^2 v^\top \boldsymbol{A}_{-1}^{-1} v}. \tag{41}$$

We next look at the residual term.

Using the formula for the OLS solution, the residual for $(x_2, y_2)$ is

$$y_2 - \hat{\theta}^\top x_2 = y_2 - y^\top \mathbf{X} (\mathbf{X}^\top \mathbf{X})^{-1} x_2. \tag{42}$$

Using the Sherman-Morrison formula, this can be written as

$$y_2 - \hat{\theta}^\top x_2 = y_2 - \left( y^\top \mathbf{X} \boldsymbol{A}_{-1}^{-1} x_2 - \frac{y^\top \mathbf{X} \boldsymbol{A}_{-1}^{-1} x_1 x_1^\top \boldsymbol{A}_{-1}^{-1} x_2}{1 + x_1^\top \boldsymbol{A}_{-1}^{-1} x_1} \right). \tag{43}$$

Substituting $x_2 = \lambda v$ and $y = \lambda c$ and combining fractions, we get

$$\begin{aligned} y_2 - \hat{\theta}^\top x_2 &= y_2 - \left( (\lambda^2 c v^\top + b_{-1}) \boldsymbol{A}_{-1}^{-1} x_2 - \frac{\lambda^2 (\lambda^2 c v^\top + b_{-1}) \boldsymbol{A}_{-1}^{-1} v v^\top \boldsymbol{A}_{-1}^{-1} x_2}{1 + \lambda^2 v^\top \boldsymbol{A}_{-1}^{-1} v} \right) \\ &= y_2 - \left( \lambda^2 c v^\top \boldsymbol{A}_{-1}^{-1} x_2 + b_{-1} \boldsymbol{A}_{-1}^{-1} x_2 - \frac{\lambda^4 c v^\top \boldsymbol{A}_{-1}^{-1} v v^\top \boldsymbol{A}_{-1}^{-1} x_2 + \lambda^2 b_{-1} \boldsymbol{A}_{-1}^{-1} v v^\top \boldsymbol{A}_{-1}^{-1} x_2}{1 + \lambda^2 v^\top \boldsymbol{A}_{-1}^{-1} v} \right). \end{aligned} \tag{44}$$

Finally, combining fractions, we get that the residual is

$$y_2 - \hat{\theta}^\top x_2 = \frac{\lambda^2 (y_2 v^\top \boldsymbol{A}_{-1}^{-1} v - c v^\top \boldsymbol{A}_{-1}^{-1} x_2 - b_{-1} \boldsymbol{A}_{-1}^{-1} x_2 v^\top \boldsymbol{A}_{-1}^{-1} v + b_{-1} \boldsymbol{A}_{-1}^{-1} v v^\top \boldsymbol{A}_{-1}^{-1} x_2) + y_2 - b_{-1} \boldsymbol{A}_{-1}^{-1} x_2}{1 + \lambda^2 v^\top \boldsymbol{A}_{-1}^{-1} v}. \tag{45}$$

Taking Equations (41) and (45) together, the influence score of $(x_2, y_2)$ is

$$e_p^\top (\mathbf{X}^\top \mathbf{X})^{-1} x_2 (y_2 - \hat{\theta}^\top x_2) = \frac{\lambda^4 s t + \lambda^2 s (y_2 - b_{-1} \boldsymbol{A}_{-1}^{-1} x_2) + \lambda^2 t e_p^\top \boldsymbol{A}_{-1}^{-1} x_2 + e_p^\top \boldsymbol{A}_{-1}^{-1} x_2 (y_2 - b_{-1} \boldsymbol{A}_{-1}^{-1} x_2)}{\lambda^4 (v^\top \boldsymbol{A}_{-1}^{-1} v)^2 + 2\lambda^2 v^\top \boldsymbol{A}_{-1}^{-1} v + 1}, \tag{46}$$

where

$$s = (v^\top \boldsymbol{A}_{-1}^{-1} v e_p^\top \boldsymbol{A}_{-1}^{-1} x_2 - e_p^\top \boldsymbol{A}_{-1}^{-1} v v^\top \boldsymbol{A}_{-1}^{-1} x_2) \tag{47}$$

$$t = (y_2 v^\top \boldsymbol{A}_{-1}^{-1} v - c v^\top \boldsymbol{A}_{-1}^{-1} x_2 - b_{-1} \boldsymbol{A}_{-1}^{-1} x_2 v^\top \boldsymbol{A}_{-1}^{-1} v + b_{-1} \boldsymbol{A}_{-1}^{-1} v v^\top \boldsymbol{A}_{-1}^{-1} x_2) \tag{48}$$

Taking a limit in Equation (46) as $\lambda \to \infty$,

$$
\begin{aligned}
\lim_{\lambda \to \infty} e_p^\top (\mathbf{X}^\top \mathbf{X})^{-1} x_2 (y_2 - \hat{\theta}^\top x_2) &= \lim_{\lambda \to \infty} \frac{\lambda^4 st + \lambda^2 s(y_2 - b_{-1} \boldsymbol{A}_{-1}^{-1} x_2) + \lambda^2 t e_p^\top \boldsymbol{A}_{-1}^{-1} x_2 + e_p^\top \boldsymbol{A}_{-1}^{-1} x_2 (y_2 - b_{-1} \boldsymbol{A}_{-1}^{-1} x_2)}{\lambda^4 (v^\top \boldsymbol{A}_{-1}^{-1} v)^2 + 2\lambda^2 v^\top \boldsymbol{A}_{-1}^{-1} v + 1} \\
&= \lim_{\lambda \to \infty} \frac{\lambda^4 st}{\lambda^4 (v^\top \boldsymbol{A}_{-1}^{-1} v)^2} \\
&= \frac{st}{(v^\top \boldsymbol{A}_{-1}^{-1} v)^2}.
\end{aligned}
\tag{49}
$$

$\square$

### C.6.2   Generality of conditions in the one-outlier theory: a simulation study.

To assess the strictness of the conditions posed in Proposition 4.3, we run a simulation study following the setup and assumptions outlined in Proposition 4.3.

Next, for $N = 1,000$, we generate 5,000 datasets for each of dimensions, $P = 3, 6,$ and 9. For each data set, we choose 1 data point uniformly at random from the inlier samples to be $(x_2, y_2)$, a random integer between 1 and $P$ (inclusive) for the value of $p$ in $e_p$, and a random unit vector for $v \in \mathbb{R}^P$. Additionally, we take $\lambda$ to be large ($10^{10}$) and choose $(x_1, y_1)$ to be the point at the first index. We then compute the values of $s$ and $t$.

In 5,000 simulations for each dimension, we observe that neither $s$ nor $t$ are ever zero. This provides further empirical evidence to suggest that Proposition 4.3 holds more broadly than just the specific toy example provided in Section 4.2.2.

### C.6.3   One-outlier example, empirical findings supplementals.

Table 12 and Table 13 display empirical findings for the data generating process described in Section 4.2. The tables present empirical evidence showing that a sufficiently far outlier will have vanishingly low influence score (see Proposition 4.3). As the black-dot point (the outlier) moves far from the group of red-cross points (the central points) in both the x and y directions, both the leverage-like term and the residual term of the influence score approach zero at rate $O(\frac{1}{\lambda})$ (see numerical results in columns 3 and 4 of Table 12). When observing the behavior of the red-cross point with the largest influence score in Table 13, we see that the leverage-like term approaches zero while the residual term stays relatively constant (within the same order of magnitude). Thus, for sufficiently large values of $(x_i, y_i)$, the influence score for the black-dot point becomes smaller than that of a red-cross point (see the highlighted values in Table 12 and Table 13). For the $(x_i, y_i)$ values with highlighted influence scores (see Table 12), both AMIP and Greedy AMIP fail (both with and without re-run). This occurs at $x = y = 10^6$ for the data generating process described in Section 4.2.

Table 12: This table shows the influence and One-Exact scores for the black-dot point at various values of $(x_i, y_i)$ for the data generating process described in Section 4.2 (see plot for the setting where $(x_i, y_i) = (10^6, 10^6)$ in Figure 1 (left)). In order to obtain the influence with respect to $\theta_1$, we let $e_1 = (0, 1)$, the standard basis vector corresponding to the x term in our linear regression setup. The influence score, $e_1^\top (\mathbf{X}^\top \mathbf{X})^{-1} x_i (y_i - \hat{\theta} x_i)$, is a product between the quantity in column 3 (which we call the leverage-like term, see Equation (3)) and column 4 (the residuals). The One-Exact score is the change in effect size that results from dropping the single data point at $(x_i, y_i)$ and refitting OLS. The influence scores highlighted in yellow are those that are smaller than the influence score of a red-cross point, leading AMIP and Greedy AMIP to misidentify the Most Influential Set of size 1, resulting in a failure with re-run.

| BLACK DOT POINT (THE OUTLIER): | | | | | |
|---|---|---|---|---|---|
| $x_i$ | $y_i$ | $e_1^\top (\mathbf{X}^\top \mathbf{X})^{-1} x_i$ | $(y_i - \hat{\theta} x_i)$ | INFLUENCE SCORE | ONE-EXACT SCORE |
| 1E1 | 1E1 | 9.36E-3 | 18.390 | 1.72E-1 | 1.90E-1 |
| 1E2 | 1E2 | 9.11E-3 | 17.981 | 1.64E-1 | 1.85 |
| 1E4 | 1E4 | 1.00E-5 | 1.97E-1 | 1.97E-5 | 2.03 |
| 1E6 | 1E6 | 1.00E-6 | 1.97E-3 | 1.97E-9 | 2.03 |
| 1E8 | 1E8 | 1.00E-8 | 2.00E-5 | 1.97E-13 | 2.18 |
| 1E10 | 1E10 | 1.00E-10 | 1.00E-5 | 9.54E-16 | 2.03 |

Table 13: This table shows the influence and One-Exact scores for the red-cross point with the largest influence score when the black-dot point (see Figure 1 (left)) is placed at the $(x_i, y_i)$ position shown in the corresponding row of Table 12. In order to obtain the influence with respect to $\theta_1$, we let $e_1 = (0, 1)$, the standard basis vector corresponding to the $x$ term in our linear regression setup. The Influence Score, $e_1^\top (\mathbf{X}^\top \mathbf{X})^{-1} x_j (y_j - \hat{\theta} x_j)$, is a product between the quantity in column 3 (which we call the leverage-like term, see Equation (3)) and column 4 (the residuals). The One-Exact score is the change in effect size that results from dropping the single data point at $(x_j, y_j)$ and refitting OLS. The influence scores highlighted in yellow are those that are larger than the influence score of the black-dot point, leading AMIP and Greedy AMIP to misidentify the Most Influential Set of size 1, resulting in a failure with re-run.

| RED CROSS POINT (A CENTRAL POINT): | | | | | |
|---|---|---|---|---|---|
| $x_j$ | $y_j$ | $e_1^\top (\mathbf{X}^\top \mathbf{X})^{-1} x_j$ | $(y_j - \hat{\theta} x_j)$ | INFLUENCE SCORE | ONE-EXACT SCORE |
| 2.13 | -0.09 | 2.02E-3 | 1.66 | 3.38E-3 | 3.40E-3 |
| -0.72 | -2.05 | 7.10E-5 | -1.57 | 1.12E-4 | 1.12E-4 |
| 2.70 | -4.85 | 7.25E-5 | -7.66 | 5.55E-7 | 5.55E-7 |
| 2.70 | -4.85 | 7.25E-8 | -7.65 | 7.63E-9 | 7.64E-9 |
| 2.70 | -4.85 | 9.97E-10 | -7.65 | 7.65E-11 | 7.66E-11 |
| 2.70 | -4.85 | 1.00E-14 | -7.65 | 7.65E-13 | 7.66E-13 |

### C.7 Multi-outlier examples

#### C.7.1 Multi-outlier failure mode theory

Additive approximations may be inaccurate, even for approximating the removal of two data points. In Proposition 4.4, we show mathematically that when a pair of points, off by a constant term, go together towards infinity, the Additive One-Exact approximation to dropping the pair tends towards zero, regardless of what the true change in effect size approaches.

**Proposition 4.4.** *Let $\lambda, c \in \mathbb{R}$. Consider a pair of data points, $(x_1, y_1) = (\lambda, \lambda)$ and $(x_2, y_2) = (\lambda, \lambda + c)$. Let $(x_n, y_n)_{n=3}^{N}$ be any points in $\mathbb{R} \times \mathbb{R}$ such that at least one of $(x_n)_{n=3}^{N}$ is non-zero. We apply OLS to the single covariate $x$ and response $y$ with no intercept; we make a decision based on the sign of the resulting effect size. As $\lambda \to \infty$, the Additive One-Exact approximation (Section 3.1) to the change in effect size from dropping $(x_1, y_1), (x_2, y_2)$ tends to zero, while the true change in effect size tends to $1 - (\sum_{n \neq 1,2}^{N} x_n y_n / \sum_{n \neq 1,2}^{N} x_n^2)$.*

*Proof.* Let $\hat{\theta}_{-1}$ denote the OLS solution fit to the data after dropping point $(x_1, y_1)$, and let $\hat{\theta}_{\{-1, -2\}}$ denote the OLS solution fit to the data after dropping the pair, $(x_1, y_1), (x_2, y_2)$. Let Add-1Exact$(1, 2)$ denote the Additive One-Exact approximation to the change in effect size after dropping $(x_1, y_1), (x_2, y_2)$. Finally, let $S_1 = \sum_{k \neq 1,2}^{N} x_n^2$ and $S_2 = \sum_{n \neq 1,2}^{N} y_n x_n$.

Add-1Exact$(1, 2)$ is expressed as

$$
\begin{aligned}
\text{Add-1Exact}(1, 2) &= (\hat{\theta} - \hat{\theta}_{-1}) + (\hat{\theta} - \hat{\theta}_{-2}) \\
&= \frac{\lambda^2(S_1 - S_2) - \lambda^3 c}{(S_1 + \lambda^2)(S_1 + 2\lambda^2)} + \frac{\lambda^2(S_1 - S_2) + \lambda^3 c}{(S_1 + \lambda^2)(S_1 + 2\lambda^2)} \\
&= \frac{2\lambda^2(S_1 - S_2)}{(S_1 + \lambda^2)(S_1 + 2\lambda^2)}.
\end{aligned}
\tag{50}
$$

As $\lambda \to \infty$, Equation (50) tends to zero.

$$
\begin{aligned}
\lim_{\lambda \to \infty} \text{Add-1Exact}(1, 2) &= \lim_{\lambda \to \infty} \frac{2(S_1 - S_2)\lambda^2}{2\lambda^4 + 3S_1\lambda^2 + S_1^2} \\
&= 0.
\end{aligned}
\tag{51}
$$

In contrast, the expression for the true change in effect size from dropping the two points is expressed as

$$
\begin{aligned}
\hat{\theta} - \hat{\theta}_{\{-1, -2\}} &= \frac{4(S_1 - S_2)\lambda^4 + 2(S_1 - S_2)S_1\lambda^2}{(S_1 + 2\lambda^2)(S_1^2 + 2S_1\lambda^2)} \\
&= \frac{4(S_1 - S_2)\lambda^4}{4S_1\lambda^4} + O(\frac{1}{\lambda^2}).
\end{aligned}
\tag{52}
$$

Taking a limit in Equation (52) as $\lambda \to \infty$,

$$
\begin{aligned}
\lim_{\lambda \to \infty} (\hat{\theta} - \hat{\theta}_{-1, -2}) &= \lim_{\lambda \to \infty} \frac{4(S_1 - S_2)\lambda^4}{4S_1\lambda^4} + O(\frac{1}{\lambda^2}) \\
&= 1 - \frac{S_2}{S_1} \\
&= 1 - \frac{\sum_{n \neq 1,2}^{N} y_n x_n}{\sum_{k \neq 1,2}^{N} x_n^2}.
\end{aligned}
\tag{53}
$$

$\square$

### C.7.2 Additional multi-outlier examples

**Adversarial Example.** Moitra & Rohatgi (2023) presents an adversarially constructed example in which there exists a small fraction of points that can be dropped such that the covariance matrix becomes singular. This leads to a failure mode of approximation algorithms. In Section 4, we attempt to alter this adversarial setup into one that might arise in natural data settings with no adversary (see Figure 1 (right)). We see that even modest levels of instability in the covariance matrix can lead to failure modes in the approximation methods.

To visualize the example presented in Moitra & Rohatgi (2023), we generate the red crosses so as to have a singular covariance matrix (see Figure 7). In particular, we generate the 1,000 red crosses with $x_n = 0$, $y_n = \epsilon_n$, and $\epsilon_n \overset{\text{iid}}{\sim} \mathcal{N}(0,1)$. We draw the 10 black dots as $x_n \overset{\text{iid}}{\sim} \mathcal{N}(-1, 0.01)$, $y_n = x_n$. When we consider both black dots and red crosses together as a single dataset, there is no poor conditioning. However, when we drop the red population, a pathological change occurs in the covariance matrix; it becomes singular. The OLS-estimated slope on the full dataset is about 1.00; dropping the black dots (1% of the data) yields a slope of exactly 0. Note, although the removal of the black-dot points do not induce a sign change in this example, going from a positive signed coefficient to 0 still constitutes a conceivable conclusion change in a data analysis.

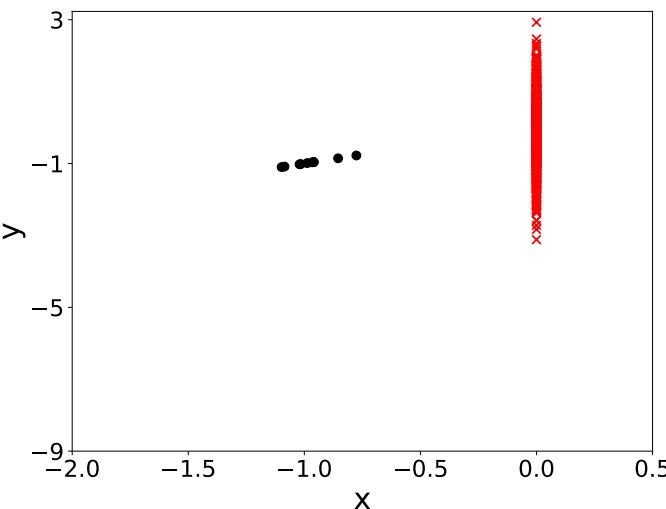

Figure 7: Example of poor conditioning presented in Section 5.1 of Moitra & Rohatgi (2023)

**Greedy AMIP Failure Example.** In the following example, we illustrate a case in which Greedy AMIP fails (See Figure 8). In particular, when there is one black dot left to remove, Greedy AMIP is unable to identify the black dot as the point to remove. This is because the residual of the last remaining black dot becomes vanishingly small when the second to last black dot is removed in the previous iteration.

In this example, we generate the 1,000 red crosses with $x_n = 0$, $y_n = \epsilon_n$, and $\epsilon_n \overset{\text{iid}}{\sim} \mathcal{N}(0,1)$. We draw the 10 black dots as $x_n \overset{\text{iid}}{\sim} \mathcal{N}(-1, 0.01)$, $y_n = -5x_n - 10$. The OLS-estimated slope on the full dataset is about 4.94; dropping the black dots (1% of the data) yields a slope of about 0.

In this example, Greedy One-Exact succeeds. For the mathematical programs algorithms, NetApprox succeeds while FH-Gurobi fails.

**Greedy AMIP and Greedy One-Exact Failure Example.** In the next example (See Figure 9), by clustering $k$ outliers tightly into a small clump, we can construct an instance where both greedy AMIP and 1sN fail to identify the $k$ outlier cluster. This repeated $k$ points centered around one clump (where $k$ is large) produces an example where the residuals are vanishingly small in the outlier cluster, while the leverage can

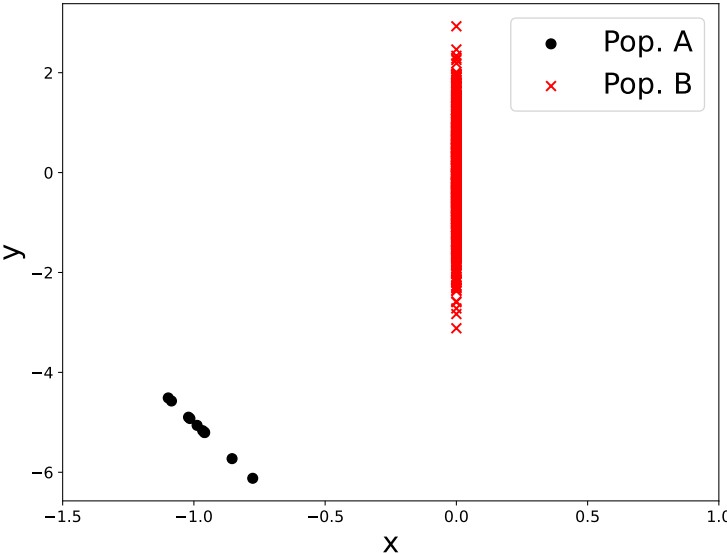

Figure 8: This is an example where Greedy AMIP fails but Greedy One-Exact succeeds.

only be as big as $1/k$. Hence, the One-Exact scores of certain points in the red inlier population (population B) will be larger. In this instance, if we computed the One-Exact score for every subset of size $k$, however, we would be able to correctly identify population A as the Most Influential Set.

For the mathematical programs algorithms, NetApprox succeeds while FH-Gurobi fails.

In this example, we generate the 1,000 red crosses with $x_n = 0$, $y_n = \epsilon_n$, and $\epsilon_n \overset{\text{iid}}{\sim} \mathcal{N}(0,1)$. We draw the 10 black dots as $x_n \overset{\text{iid}}{\sim} \mathcal{N}(-1, 10^{-7})$, $y_n = -5x_n - 10$. The OLS-estimated slope on the full dataset is about 4.94; dropping the black dots (1% of the data) yields a slope of about 0.

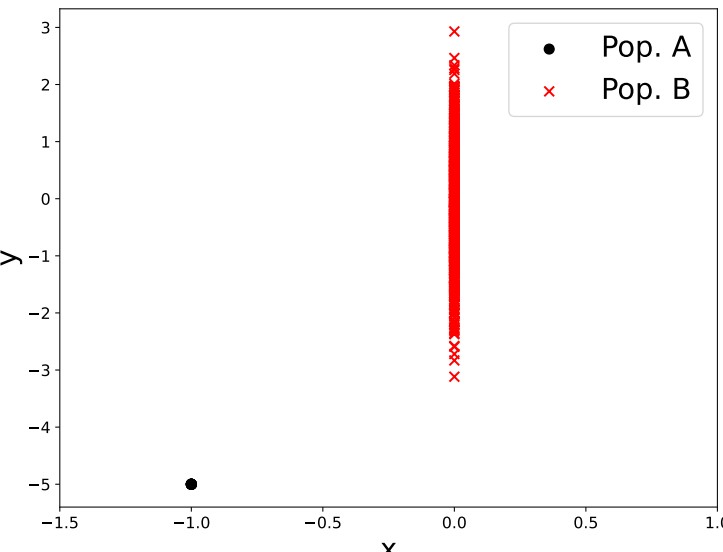

Figure 9: This is an example where Greedy AMIP and Greedy One-Exact both fail.

Observe from Equation (18) that the error arises from a failure to correctly reweight the inverse Hessian term by the dropped subset, $S$. While AMIP disregards this reweighting entirely, Additive One-Exact decreases the error by reweighting the Hessian on an individual point basis (See Equation (19)).

