# OpenReview forum: "Approximations to worst-case data dropping: unmasking failure modes"
_TMLR — Accepted by TMLR_

### Review · Reviewer_sA6M · 2025-05-04

**Summary Of Contributions:**

This paper addresses the issue of model generalization sensitivity in the context of OLS regression. Specifically, it examines the non-robustness of model conclusions to the removal of a small fraction of data points—a problem that is computationally intractable to check exactly. While several approximation methods have been proposed to detect such non-robustness, the authors demonstrate that many of these fail in practice, even in the relatively simple OLS setting. Through empirical evaluation, they show that a recursive greedy algorithm not only reliably detects non-robustness but also does so with significantly greater computational efficiency than competing methods.

**Audience:**

Yes

**Claims And Evidence:**

Yes

**Requested Changes:**

Please see weaknesses.

**Strengths And Weaknesses:**

Strengths

- The paper provides a comprehensive discussion and analysis of relevant related work.
- Comparisons are clearly presented from both theoretical and empirical perspectives.


Weaknesses

- The paper states to focus on realistic data configurations, but the reviewer could not find any experiments conducted on real-world datasets. More comprehensive evaluations using several real-world datasets would be beneficial.

- The paper appears to be a review of recent studies on data dropping, without presenting any particularly novel findings. As such, the reviewer feels that the contribution is somewhat limited in terms of originality.

- Additionally, the work focuses primarily on OLS, which is relatively simple. The reviewer is concerned about the practical relevance and applicability of this work.

---

> ### Author Response · Authors · 2025-05-21
> **Response from the authors**
>
> We are glad to hear that the reviewer found our theoretical and empirical comparisons to be clear and our treatment of related work to be  comprehensive. We appreciate the constructive comments and suggestions, and we address them in turn below.
>
> ---
>
> ## **Comprehensive Evaluations using Several Real-World Data Sets**
>
> We agree with the reviewer that demonstrating practical relevance via real-world data sets is critical. In response, we have added **comprehensive evaluations on several real-world data sets**—from genomics, animal ecology, forest ecology, plant physiology, and real-estate markets—to complement our synthetic examples. These can be found as new sections, **Section 5** and **Appendix D.4**, in our manuscript.
>
> ---
>
> ## **Highlighting our Contributions**
>
> We discuss the originality of our contributions in *General Reply: Rebuttal summary and changes in the new version of the paper (Part I)* and continue this discussion below:
>
> Recall that we are interested in a particular form of robustness introduced by Broderick et al. 2020: whether it is possible to drop at most $\alpha$ fraction of data points in a data analysis (for $\alpha$ < 1%) and change the conclusions of the analysis. **We provide the first systematic comparison of multiple approximation methods at the task of detecting this form of non-robustness.** We provide novel theory and experiments to support our conclusions. After adding additional tests in response to reviewers, **we find that only a single approximation method (out of seven total methods) succeeds across all of our experiments.**
>
> For instance, in contrast to Kuschnig et al. (2021) we consider failures to be cases where the approximation comes to a different conclusion on non-robustness than the true non-robustness, rather than changes of magnitude.,  Moreover, unlike Kuschnig et al. (2021), we provide comparisons to the OLS-specific approximations. Analogously, Hu et al. (2024), Moitra and Rohatgi (2023), Freund and Hopkins (2023), Rubinstein and Hopkins (2025) are all interested in exact recovery of the most influential subset. (In fact, the latter four are only interested in the size of the most influential subset rather than its elements.) As we argue in the paper, there can be cases of practical interest where the most influential subset isn’t perfectly identified but non-robustness remains the same.
>
> The final two major forms of **new contributions** are:
>
> - We are the first to surface a series of **targeted failure mode examples** for this type of non-robustness—both in realistic synthetic and real-world data—that are deliberately simple yet reveal fundamental flaws in the existing data-dropping approximations.
> - We provide **new theoretical results** that suggest failure modes of AMIP with extreme outliers and masking:
>    - **Proposition 4.3**: Behavior of influence scores as an outlier point is taken far away from the rest of the points.
>    - **Proposition 4.4**: A formal illustration of non-additivity (e.g., a demonstration of masking) for a pair of data points.
>
> ---
>
> ## **Rationale for Focusing on OLS**
>
> OLS remains one of the most widely used tools in scientific and economic research. We have added a new discussion in the manuscript (**see blue text on page 3**) that articulates the importance of the sign of an effect size in linear regression, in decision-making across scientific and economic studies (see **Section 2: Justification for considering sign change under the General Reply**). We summarize the results below, for your convenience:
>
> “In many real-world settings, the direction of an estimated effect—whether positive or negative—can alter interpretations and policy actions. For example, in “Contradicted and Initially Stronger Effects in Highly Cited Clinical Research,” Ioannidis (2005) highlights prominent medical studies in which the direction of an estimated effect was later reversed by subsequent research; that is, the reversal represented a different conclusion about a clinical treatment from the original study [11]. Similarly, in Mostly Harmless Econometrics, Angrist & Pischke (2009, Section 4.1.2) demonstrate the importance of the direction of causal effects in shaping policy-relevant inferences [12].”
>
> Furthermore, by surfacing failure modes in OLS, we make a broader point that, if existing methods are unreliable in OLS, extending them to more complex models may result in similar issues. Finally, many recent robustness approximation methods—such as NetApprox (Moitra and Rohatgi, 2023 [2]) and both variants of FH-Gurobi (Freund and Hopkins, 2023 [3])—are currently designed only for OLS. It is for these reasons that we choose to focus on OLS.

---

### Review · Reviewer_MPJ3 · 2025-05-04

**Summary Of Contributions:**

Authors consider the following question: Given a dataset consisting $X \in \mathbb{R}^{N \times P}$ and $y \in \mathbb{R}^N$, $p \in [P]$ and $\alpha \leq 0.01$ are the $\alpha N$ or less rows in $X$ such that dropping these rows causes a sign flip in the $p$-th coordinate of the optimum solution of the OLS estimator.

They compare the following algorithms:
additive approximations: Approximate Maximum Influence Perturbation (AMIP) and Additive One-Exact
Greedy approximations: Greedy One-Exact and Greedy AMIP
Approximations Specific to Ordinary Least Squares: NetApprox, FH-Gurobi (with and without warm start)

More precisely they construct three synthetic datasets (a daataset with a  single outlier and two datasets with multiple outliers) and analyze the running time of the algorithms as well as whether they are able to answer the question correctly using the output of the algorithms.

Their result is Greedy One-Exact,NetApprox and FH-Gurobi (with warm start) answer the question correctly for all three datasets while each other algorithm fails for at least one dataset. Moreover the additvie approximationalgorithm run faster than the greedy algorithms which run faster as the specific algorithms. Authors thus recommend to use greedy algorithms if there is enough compuetation power.

They further enhance their findings by stating and proving why the algorithms fail on these instances.

**Audience:**

Yes

**Claims And Evidence:**

Yes

**Requested Changes:**

- There could be an explanation on how $p$ is chosen.

- There shouild be a reference or derivation of equation (3)

**Strengths And Weaknesses:**

Strengths:

- write up seems solid. Experiemnts and theoetic reasoning are provided
- in general the problem whether dropping a small amount of the data leads to significant change seems to be an interesting question

Weaknesses:

- only three synthetic datasets are considered.
- the criteria (i.e. change of sign a specific coordinate) seems a bit arbitrary.

---

> ### Author Response · Authors · 2025-05-21
> **Response from the authors**
>
> We thank the reviewer for their positive assessment of our write-up and their interest in the question of robustness to data dropping. We appreciate the reviewer’s concerns and are pleased to address them below.
>
> ---
>
> ## **Addressing Reviewer Concerns on the Diversity of Datasets**
>
> We agree that testing on real-world data examples enhances our claims. In response, we have added **several real-world data sets**—from genomics, animal ecology, forest ecology, plant physiology, and real-estate markets—to complement our synthetic examples. These have been added as new sections, **Section 5** and **Appendix D.4**, in the manuscript. These data sets help demonstrate that the types of failures we study indeed arise in a variety of common applied settings.
>
> ---
>
> ## **Addressing Reviewer Concern on Using Sign Change as the Criterion**
>
> The sign of an estimated effect often carries interpretive weight, in many fields—such as biomedicine and economics—where the direction of an effect estimate impacts how results are interpreted and acted upon. For example, in “Contradicted and Initially Stronger Effects in Highly Cited Clinical Research,” Ioannidis (2005) highlights prominent medical studies where the direction of the estimated effect was later reversed by subsequent research, leading to fundamentally different conclusions [11]. Similarly, in Mostly Harmless Econometrics (Section 4.1.2), Angrist and Pischke (2009) highlight the importance of understanding the direction of causal effects in shaping policy-relevant inferences [12]. For this reason, sign robustness is an important consideration. Thus, we follow precedent from prior work in this space [Broderick et al. 2020 [1], Moitra & Rohatgi 2023 [2], Freund and Hopkins 2023 [3], Rubinstein and Hopkins 2025 [4]], where sign change is used as an indicator of conclusion change. We have added further discussion of this in the revised paper---this can be found in the **blue text on page 3 of the manuscript**.
>
> ---
>
> ## **Clarifying that the Choice of Index $p$ is User-specified**
>
> We clarify that the dimension p corresponds to the coefficient of interest and is user-specified. For instance, in evaluating the robustness of a treatment effect in a randomized trial, the researcher would set p to the index corresponding to the treatment variable. We have added text clarifying that p depends on the quantity of interest to a user in **Section 2** **(see blue text on page 3 of the manuscript)**.
>
> ---
>
> ## **Adding Derivation of Equation (3)**
>
> We have added a complete derivation of the influence score used in Equation (3) in **Appendix C.1** and now reference this derivation directly in the main text for clarity **(see blue text at the top of page 4 of the manuscript)**.

---

### Review · Reviewer_Y1Kf · 2025-05-07

**Summary Of Contributions:**

This paper investigates the robustness of data-dropping methods in the context
of ordinary least squares (OLS) for linear regression. The authors focus on
Approximate Maximum Influence Perturbation (AMIP) and compare various approaches
that utilize AMIP to detect the most influential sets of data points.  They
demonstrate that many of these approaches can fail to detect the most
influential sets of data points in simplified data scenarios. The authors also
provide theoretical insights into why AMIP may not serve as a reliable criterion
in certain situations.

**Audience:**

Yes

**Claims And Evidence:**

Yes

**Requested Changes:**

1. The weaknesses mentioned above should be addressed.
2. Put the expression after $max_{w\in W_α}$ in parentheses, or move
   $max_{w\in W_α}$ in front of $\hat{θ}_p(w)$ in (1).
3. Equation (3) already requires that $N>p$ and $X$ is full rank. It is better
   to clarify this point here rather than later in the paper.
4. I am not clear what you mean by "Analysis represents the cost of the data
   analysis" on page 5. Is this just the OLS calculation?
5. For the Additive One-Exact method, does it only require finding the data
   point with the largest influence? If so, should $N\log N$ be replaced by
   $N$, as this can be done in linear time? This is in the paragraph above
   Section 3.2.
6. The last paragraph on page 6 talks about the equivalence of two failures,
   but cases (1) and (2) refer to successes. This can be revised for easier
   reading.
7. Above Section 4.2, by "overestimation", do you mean a false positive—i.e., the
   method detects non-robustness when it should not? Please clarify this point.
8. The numerical experiments fit OLS with an intercept, while the theoretical
   results fit without an intercept. Although it is mentioned elsewhere that the
   numerical results are not much affected by the intercept, this inconsistency
   may confuse readers. Please make them consistent and state that the intercept
   does not significantly affect the numerical results.

**Strengths And Weaknesses:**

Strengths:
1. The paper is overall well-written and easy to follow. The authors provide a
   clear motivation for the study, and the presentation is clear and coherent.
2. The paper addresses an important problem in statistical analysis and data
   science. It may have a significant impact on real-life decision making if the
   conclusion from the data analysis can be reversed by removing even a tiny
   proportion of the data.
3. The theoretical insights provided in the paper match the empirical
   observations well.

Weaknesses:
1. The scenarios and numerical examples considered are simple. Real data
   patterns are often more complex. Not requesting to investigate more
   complicate cased, but it would be better to make this point clear.
2. According to the theoretical results, the major reason for not being able to
   detect the impact of the outliers is due to the limitations of AMIP. It's
   better to clarify this point. In real data analysis, it is very important to
   perform regression diagnostics to identify outliers, and the outliers in the
   paper can be easily identified by classical diagnostic techniques.

---

> ### Author Response · Authors · 2025-05-21
> **Response from the authors (Part I)**
>
> We thank the reviewer for the positive assessment of the paper’s clarity, motivation, and relevance. We particularly appreciate the recognition that detecting robustness failures—especially those due to removing a very small fraction of the data—can have significant real-world impact.
>
> Below, we address each of the reviewer’s thoughtful concerns.
>
> ---
>
> ## **Addressing Reviewer Concerns on Real-World Data Complexity**
>
> We have added a series of analyses of real data sets from a diverse range of application domains including genomics, animal ecology, forest ecology, plant physiology, and real-estate markets to demonstrate our failure modes (see **Section 5** and **Appendix D.4** in the manuscript). We found the real-world examples to be an especially valuable addition, as the added complexity of real-world data helped surface limitations (in particular, see the new failure modes of FH-Gurobi (warm-start) and NetApprox on **page 12 and 13** of the manuscript, respectively) that were not surfaced in synthetic examples alone.
>
> ---
>
> ## **Clarifying the Role of AMIP and Discussing the Importance of Regression Diagnostics**
>
> We agree that classical regression diagnostics are essential. We have revised the discussion section (see added discussion at the **top of page 15** in **Section 7**) to clarify that our surfaced failures reflect the limitations of current approximations (such as AMIP) rather than the failure of regression analysis per se. We also underscore the importance of combining robustness approximations with visual inspection and diagnostic tools in practice. While the approximation failures we surface in the present manuscript manifest as clearly visible gross outliers, we note that previous works have identified many real-life cases where existing approximations work well, and these include data-dropping non-robust cases without gross outliers [Broderick et al., 2020] [1]. In these cases, a simple visualization or other classical regression diagnostics might not as easily detect the non-robustness.
>
> ---
>
> ## **Clarifying the Runtime for Additive One-Exact**
>
> The reviewer asks if the Additive One-Exact method “only require[s] finding the data point with the largest influence.” We first answer the question directly and then also answer a different interpretation of the question.
>
> The Additive One-Exact method requires finding the $\lfloor \alpha N \rfloor$ data points with the largest exact impact (rather than only the top point) because it uses a sum of individual impacts to approximate the impact of leaving out the worst-case group (the Most Influential Subset). Thus, this step cannot be done in linear time. However, it is true that finding the top $\lfloor \alpha N \rfloor$ points in a list can be done in time $N \log  (\lfloor \alpha N \rfloor) $ with a heap, and since it is often the case that $\lfloor \alpha N \rfloor << N$, we have made this change in runtime in **Section 3** of the manuscript.
>
> However, we suspect that the reviewer may have been asking about the runtime for Greedy One-Exact. If this is the case, then it is correct to point out that, at each single iteration of greedy, we do only need to identify the point with the top score and thus do not need to sort the full list when performing the greedy algorithm update. We have updated the manuscript (see the blue text in **Section 3**) to remove the $\log N$ factor for Greedy One-Exact that erroneously appeared before.
>
> ---
>
> ## **Addressing Reviewer Concerns on the Effect of the Intercept Term in Numerical Experiments**
>
> We have added intercept-free numerical experiments to **Appendix D.4.2** showing that the intercept does not significantly affect the numerical results. We also included a more nuanced discussion of the inclusion of the intercept term.
>
> (response continued in part II)

---

> > ### Author Response · Authors · 2025-05-21
> > **Response from the authors (Part II)**
> >
> > (response continued from part I)
> >
> > ---
> >
> > ## Notational Clarity and Minor Edits
> > We appreciate the detailed suggestions. We have implemented the following:
> >
> > - **Q1: Put the expression after $\max_{w \in W_{\alpha}}$  in parentheses, or move $\max_{w \in W_{\alpha}}$  in front of $\hat{\theta}_p(w)$ in (1).**
> >
> > A1: We have now placed parentheses around the objective in Equation (1).
> >
> > - **Q2: Equation (3) already requires that $N>p$ and $X$ is full rank. It is better to clarify this point here rather than later in the paper.**
> >
> > A2: We have now clarified these points where Equation (3) is introduced.
> >
> > - **Q3: I am not clear what you mean by “Analysis represents the cost of the data analysis" on page 5. Is this just the OLS calculation?**
> >
> > A3: The text string “Analysis” denotes the cost of running a general data analysis. We include theoretical runtimes for a generic data analyses (and use this notation) because both Additive and Greedy One-Exact can be extended beyond OLS to any generic data analysis and both Additive and Greedy AMIP to any setting in which influence functions can be computed (e.g., Z-estimators).
> >
> > - Q4: **The last paragraph on page 6 talks about the equivalence of two failures, but cases (1) and (2) refer to successes. This can be revised for easier reading.**
> >
> > A4: We have removed this portion of the text and have re-written it to more clearly remark on the equivalence between the failure types for FH-Gurobi. The new text can be found **in blue under Section 4.1**.
> >
> > - Q5: **Above Section 4.2, by “overestimation,” do you mean a false positive—i.e., the method detects non-robustness when it should not? Please clarify this point.**
> >
> > A5: The nice thing about these approximations is that, by design, they don’t return false positives so long as the user is willing to re-run their data analysis once without the identified subset. So yes, we do mean false positives here. If re-running without the dropped points results in a conclusion change, then the non-robustness is conclusive. If the re-run does not result in a conclusion change, then the approximation reports that no conclusion change has occurred. Thus, we are motivated by scenarios where non-robustness goes undetected (i.e., false negatives). We have added this clarification on **the bottom of Section 4.1** of the manuscript.

---

### Author Response · Authors · 2025-05-21
**General Reply:  Rebuttal summary and changes in the new version of the paper (Part I)**

We thank the reviewers for all their valuable feedback and constructive suggestions. We are encouraged that the reviewers found our problem setup, empirical perspectives, and theoretical insights relevant and meaningful for real-world data analyses. We begin by summarizing the main contribution of our paper.

Recall that we are interested in a particular form of robustness introduced by Broderick et al. (2020): whether it is possible to drop at most $\alpha$ fraction of data points in a data analysis (for $\alpha$ < 1%) and change the conclusions of the analysis. We provide the first systematic comparison of multiple approximation methods at the task of detecting this form of non-robustness. We provide novel theory and experiments to support our conclusions. After adding additional tests in response to reviewers, we find only a single approximation method (out of seven total methods) succeeds across all of our experiments.

In response to the concerns raised, we have revised the manuscript substantially. We have added new content to strengthen the practical relevance of our findings, justify our design choices, and more closely tie our theory to our numerical findings. We carefully address each reviewer’s concerns in our individual comments to reviewers. We here summarize our major changes to the manuscript in response to the reviewers’ valuable feedback:

---

# Added Sections and Appendices:

## **Section 5: Real-world data case studies**
To illustrate the practical relevance of our findings, we added a series of real-world failure mode examples:

### **Single-cell Genomics (Multi-outlier example)**

- **Dataset:** 65,539 single-cell observations measuring gene expression in the mouse visual cortex. Data set is highly zero-inflated.
- **Setup:** OLS regression (with intercept) of gene expressions Gad1 on Vip.
- **Non-robust:** Dropping just 172 points (0.26%) flips the regression coefficient from +0.536 to –0.003.
- **Failures:** AMIP and Additive One-Exact fail with re-run. FH-Gurobi returns too large a subset (>1%), and FH-Gurobi (warm-start) does not identify a small subset that can change the sign.

### **Ames Housing (Multi-outlier example)**

- **Dataset:** Subsample of housing data from Ames, Iowa. When considering the Pool Area regressor, the data are highly zero-inflated (most houses do not have pools).
- **Setup:** OLS regression (with intercept) of SalePrice on PoolArea.
- **Non-robust:** Dropping 3 points changes coefficient from +207.78 to –6.67.
- **Failures:** AMIP and Additive One-Exact fail with re-run, predicting no sign change even upon dropping the subset that the algorithm chooses.

### **Bird Morphometrics (One-outlier example)**

- **Dataset:** 1,295 observations on various features of the saltmarsh sparrow.
- **Setup:** Multivariate OLS of tarsus length on head, wing, beak length measurements.
- **Non-robust:** Removing just one observation changes the head-length coefficient from –0.69 to +0.399.
- **Failures:** NetApprox and both FH-Gurobi variants fail to identify the correct point or return subsets >1%.

---

## **Appendix B.4: Case influence analysis**
We added a section distinguishing our work from the case influence analysis literature, which studies the importance of a non-worst case group of data points on whole-model sensitivity (e.g., a posterior mean), rather than a worst-case group on the sensitivity of a particular inferential quantity (e.g., a particular OLS regression coefficient).

---

## **Appendix D.2: Failure modes without an intercept term**
We re-ran our experiments without intercept terms to show that the intercept has negligible impact on most numerical results.

---

## **Appendix D.3: Failure with and without re-run for FH-Gurobi**
In this section, we discuss the distinction between failure with and without re-run for FH-Gurobi.

---

## **Appendix D.4: Successful examples in real-world data**
To get a sense of when these data-dropping approximations do succeed in the real world, we provide two real-world data examples (respectively concerning forestry and photosynthesis) where the analyses are non-robust to small-fraction data dropping, yet for which all methods succeed.

---

### Author Response · Authors · 2025-05-21
**General Reply:  Rebuttal summary and changes in the new version of the paper (Part II)**

# **Revised Sections and Appendices:**


## **Abstract: Revisions reflecting new results**
We updated the abstract to reflect the new results after adding in the experiments on real-world data.

---

## **Section 2: Justification for considering sign change**
In addition to our previous justifications for focusing on sign change as our criterion, we now provide additional motivation for why sign change might be an important quantity of interest in practical settings (see blue text on page 3).

---

## **Section 3 + Section C.4: Runtime corrections**
We corrected the runtime complexity of the greedy algorithms and clarified the runtime complexity of the additive algorithms.

---

## **Section 7: Discussion**
We now explicitly acknowledge that real data often exhibit more complexity than our synthetic setups. We also clarified that our failures reflect limitations of the approximation methods—not regression analysis itself—and that these tools should be paired with standard diagnostic practices.

---

Once again, we appreciate the constructive feedback the reviewers have provided. We would be happy to hear any further suggestions that may help to improve our work.

Best,

Authors

---

> ### Author Response · Authors · 2025-05-21
> **References for Reviewer Responses**
>
> Below, we give the citations that are referenced throughout our responses.
>
> ---
>
> # Works Cited.
>
> [1] Tamara Broderick, Ryan Giordano, and Rachael Meager. An automatic finite-sample robustness metric: When can dropping a little data make a big difference? arXiv preprint arXiv:2011.14999v1, 2020.
>
> [2] Ankur Moitra and Dhruv Rohatgi. Provably auditing ordinary least squares in low dimensions. The 11th International Conference on Learning Representations (ICLR 2023), 2023.
>
> [3] Daniel Freund and Samuel Hopkins. Towards practical robustness auditing for linear regression. arXiv preprint arXiv:2307.16315v1, 2023.
>
> [4] Ittai Rubinstein and Samuel Hopkins. Robustness auditing for linear regression: To singularity and beyond. The 13th International Conference on Learning Representations (ICLR 2025), 2025.
>
> [5] Tin D. Nguyen, Ryan Giordano, Rachael Meager, and Tamara Broderick. Using gradients to
> check sensitivity of MCMC-based analyses to removing data. In ICML 2024 Workshop on Differentiable Almost Everything: Differentiable Relaxations, Algorithms, Operators, and Simulators, 2024.
>
> [6] Manuela Angelucci and Giacomo De Giorgi. Indirect effects of an aid program: how do cash transfers affect
> ineligibles’ consumption? American Economic Review, 99(1):486–508, 2009.
>
> [7] Amy Finkelstein, Sarah Taubman, Bill Wright, Mira Bernstein, Jonathan Gruber, Joseph P. Newhouse, Heidi
> Allen, Katherine Baicker, and the Oregon Health Study Group. The Oregon health insurance experiment:
> evidence from the first year. The Quarterly Journal of Economics, 127(3):1057–1106, 2012.
>
> [8] Michael Di and Ke Xu. Covid-19 vaccine and post-pandemic recovery: Evidence from bitcoin cross-asset
> implied volatility spillover. Finance Research Letters, 50:103289, 2022.
>
> [9] Yuzheng Hu, Pingbang Hu, Han Zhao, and Jiaqi W. Ma. Most influential subset selection: Challenges,
> promises, and beyond. In Advances in Neural Information Processing Systems, volume 38, 2024.
>
> [10] Nikolas Kuschnig, Gregor Zens, and Jesús C. Cuaresma. Hidden in plain sight: Influential sets in linear
> models. Technical report, Vienna University of Economics and Business, 2021. CESifo Working Paper. doi: 10.2139/ssrn.3819102.
>
> [11] Ioannidis JP. Contradicted and initially stronger effects in highly cited clinical research. JAMA. 2005 Jul 13;294(2):218-28. doi: 10.1001/jama.294.2.218. PMID: 16014596.
>
> [12] Angrist, Joshua D., and Jörn-Steffen Pischke. Mostly harmless econometrics: An empiricist's companion. Princeton university press, 2009.

---

### Decision · Action_Editor_arGQ · 2025-06-06

**Recommendation:** Accept as is

**Audience:**

Yes

**Audience Explanation:**

Yes, this paper is clearly in scope, and all reviewers agree.

**Claims And Evidence:**

Yes

**Claims Explanation:**

All reviewers agree that claims are supported by evidence in the paper.  There were some small concerns, and all were addressed by the authors.